# On Convergence of FedProx: Local Dissimilarity Invariant Bounds, Non-smoothness and Beyond

**Xiao-Tong Yuan**
Nanjing Univ. of Information Sci. & Tech.
219 Ningliu Rd, Nanjing, China
xtyuan1980@gmail.com

**Ping Li**
LinkedIn Ads
700 Bellevue Way NE, Bellevue, WA, USA
pinli@linkedin.com

## Abstract

The `FedProx` algorithm is a simple yet powerful distributed proximal point optimization method widely used for federated learning (FL) over heterogeneous data. Despite its popularity and remarkable success witnessed in practice, the theoretical understanding of FedProx is largely underinvestigated: the appealing convergence behavior of `FedProx` is so far characterized under certain non-standard and unrealistic dissimilarity assumptions of local functions, and the results are limited to smooth optimization problems. In order to remedy these deficiencies, we develop a novel local dissimilarity invariant convergence theory for `FedProx` and its minibatch stochastic extension through the lens of algorithmic stability. As a result, we contribute to derive several new and deeper insights into `FedProx` for non-convex federated optimization including: 1) convergence guarantees invariant to certain stringent local dissimilarity conditions; 2) convergence guarantees for non-smooth FL problems; and 3) linear speedup with respect to size of minibatch and number of sampled devices. Our theory for the first time reveals that local dissimilarity and smoothness are not must-have for `FedProx` to get favorable complexity bounds.

## 1  Introduction

Federated Learning (FL) has recently emerged as a promising paradigm for communication-efficient distributed learning on remote devices, such as smartphones, internet of things, or agents (Konečný et al., 2016; Yang et al., 2019). The goal of FL is to collaboratively train a shared model that works favorably for all the local data but without requiring the learners to transmit raw data across the network. The principle of optimizing a global model while keeping data localized can be beneficial for both computational efficiency and data privacy (Bhowmick et al., 2018). While resembling the classic distributed learning regimes, there are two most distinct features associated with FL: 1) large statistical heterogeneity of local data mainly due to the non-iid manner of data generalization and collection across the devices (Hard et al., 2020); and 2) partial participation of devices in the network mainly due to the massive number of devices. These fundamental challenges make FL highly demanding to tackle, both in terms of optimization algorithm design and in terms of theoretical understanding of convergence behavior (Li et al., 2020a).

FL is most conventionally formulated as the following problem of global population risk minimization averaged over a set of $M$ devices:

$$\min_{w \in \mathbb{R}^p} \bar{R}(w) := \frac{1}{M} \sum_{m=1}^{M} \left\{ R^{(m)}(w) := \mathbb{E}_{Z^{(m)} \sim \mathcal{D}^{(m)}}[\ell^{(m)}(w; Z^{(m)})] \right\}, \tag{1}$$

where $R^{(m)}$ is the local population risk on device $m$, $\ell^{(m)} : \mathbb{R}^p \times \mathcal{Z}^{(m)} \mapsto R^+$ is a non-negative loss function whose value $\ell^{(m)}(w; Z^{(m)})$ measures the loss over a random data point $Z^{(m)} \in \mathcal{Z}^{(m)}$ with parameter $w$, $\mathcal{D}^{(m)}$ represents an underlying random data distribution over $\mathcal{Z}^{(m)}$. Since the data distribution is typically unknown, the following empirical risk minimization (ERM) version of (1) is

often considered alternatively:

$$\min_{w \in \mathbb{R}^p} \bar{R}_{\texttt{erm}}(w) := \frac{1}{M} \sum_{m=1}^{M} \left\{ R_{\texttt{erm}}^{(m)}(w) := \frac{1}{N_m} \sum_{i=1}^{N_m} \ell^{(m)}(w; z_i^{(m)}) \right\}, \tag{2}$$

where $R_{\texttt{erm}}^{(m)}$ is the local empirical risk over the training sample $D^{(m)} = \{z_i^{(m)}\}_{i=1}^{N_m}$ on device $m$. The sample size $N_m$ may vary significantly across devices, which can be regarded as another source of data heterogeneity. Federated optimization algorithms for solving (1) or (2) have attracted significant research interest from both academia and industry, with a rich body of efficient solutions developed that can flexibly adapt to the communication-computation tradeoffs and data/system heterogeneity. Several popularly used FL algorithms for this setting include FedAvg (McMahan et al., 2017), FedProx (Li et al., 2020b), SCAFFOLD (Karimireddy et al., 2020), and FedPD (Zhang et al., 2020), to name a few. A consensus among these methods on communication-efficient implementation is trying to extensively update the local models (e.g., with plenty epochs of local optimization) over subsets of devices so as to quickly find an optimal global model using a minimal number of inter-device communication rounds for model aggregation.

In this paper, we revisit the FedProx algorithm which is one of the most prominent frameworks for heterogeneous federated optimization. Reasons for the interests of FedProx include implementation simplicity, low communication cost, promise in dealing with data heterogeneity and tolerance to partial participation of devices (Li et al., 2020b). We analyze its convergence behavior, expose problems, and propose alternatives more suitable for scaling up and generalization. We contribute to derive several new and deeper theoretical insights into the algorithm from a novel perspective of algorithmic stability and generalization theory.

## 1.1 Review of FedProx

For solving FL problems in the presence of data heterogeneity, methods such as FedAvg based on local stochastic gradient descent (SGD) can fail to converge in practice when the selected devices perform too many local updates (Li et al., 2020b). To mitigate this issue, FedProx (Li et al., 2020b) was recently proposed for solving the empirical FL problem (2) using the (inexact) proximal point update for local optimization. The benefits of FedProx include: 1) it provides more stable local updates by explicitly enforcing the local optimization in the vicinity of the global model to date; 2) the method comes with convergence guarantees for both convex and non-convex functions, even under partial participation and very dissimilar amounts of local updates (Li et al., 2020a). More specifically, at each time instance $t$, FedProx uniformly randomly selects a subset $I_t \subseteq [M]$ of devices and introduces for each device $\xi \in I_t$ the following proximal point ERM sub-problem for local update around the previous global model $w_{t-1}$:

$$w_t^{(\xi)} \approx \arg\min_{w \in \mathbb{R}^p} \left\{ Q_{\texttt{erm}}^{(\xi)}(w; w_{t-1}) := R_{\texttt{erm}}^{(\xi)}(w) + \frac{1}{2\eta_t} \|w - w_{t-1}\|^2 \right\}, \tag{3}$$

where $\eta_t > 0$ is the learning rate that controls the impact of the proximal term. Then the global model is updated by uniformly aggregating those local updates from $I_t$ as

$$w_t = \frac{1}{|I_t|} \sum_{\xi \in I_t} w_t^{(\xi)}.$$

In the extreme case of allowing $\eta_t \to +\infty$ in (3), FedProx reduces to the regime of FedAvg if using SGD for local optimization. Since its inception, FedProx and its variants have received significant interests in research (Pathak and Wainwright, 2020; Nguyen et al., 2020; Li et al., 2019) and become an algorithm of choice in application areas such as automatous driving (Donevski et al., 2021) and computer vision (He et al., 2021). Theoretically, FedProx comes with convergence guarantees under the following bounded *local gradient dissimilarity* assumption that captures the statistical heterogeneity of local objectives across the network:

**Definition 1** (($B, H$)-LGD). *We say the local functions $R^{(m)}$ have $(B, H)$-local gradient dissimilarity (LGD) if the following holds for all $w \in \mathbb{R}^p$:*

$$\frac{1}{M} \sum_{m=1}^{M} \|\nabla R^{(m)}(w)\|^2 \leq B^2 \|\nabla \bar{R}(w)\|^2 + H^2.$$

The above definition naturally extends to the local empirical risks $\{R_{\text{erm}}^{(m)}\}_{m=1}^M$. Specially in the homogenous setting where $R^{(m)} \equiv \bar{R}, \forall m \in [M]$, we have $B = 1$ and $H = 0$. Under $(B, 0)$-LGD and some regularization conditions on the modulus $B$, it was shown that `FedProx` for non-convex problems requires $T = \mathcal{O}\left(\frac{1}{\epsilon}\right)$ rounds of inter-device communication to reach an $\epsilon$-stationary solution in the sense of $\frac{1}{T}\sum_{t=1}^T \|\nabla \bar{R}_{\text{erm}}(w_t)\|^2 \le \epsilon$ (Li et al., 2020b). Similar guarantees have also been established for a variant of `FedProx` with non-uniform model aggregation (Nguyen et al., 2020).

*Open issues and motivation.* In spite of the remarkable success achieved by `FedProx` and its variants, there are still a number of important theoretical issues regarding the unrealistic assumptions, restrictive problem regimes and expensive local oracle cost that remain open for exploration, as specified below.

- **Local dissimilarity condition**. The appealing convergence behavior of `FedProx` is so far characterized under a key but non-standard $(B, 0)$-LGD condition (cf. Definition 1) with $B > 0$. Such a condition is obviously unrealistic in practice: it essentially requires the local objectives share the same stationary point as the global objective since $\|\nabla \bar{R}_{\text{erm}}(w)\| = 0$ implies $\|\nabla R_{\text{erm}}^{(m)}(w)\| = 0$ for all $m \in [M]$. However, if the optima of $R_{\text{erm}}^{(m)}$ are exactly (or even approximately) the same, there would be little point in distributing data across devices for federated learning. *It is thus desirable to understand the convergence behavior of `FedProx` for heterogeneous FL without imposing stringent local dissimilarity conditions like $(B, 0)$-LGD.*

- **Non-smooth optimization**. The existing convergence guarantees of `FedProx` are only available for FL with smooth losses. More often than not, however, FL applications involve non-smooth objectives due to the popularity of non-smooth losses (e.g., hinge loss and absolute loss) in machine learning, and training deep neural networks with non-smooth activation like ReLU. *Therefore, it is desirable to understand the convergence behavior of `FedProx` in non-smooth problem regimes.*

- **Local oracle complexity**. Unlike the (stochastic) first-order oracles such as SGD used by `FedAvg`, the proximal point oracle (3) for local update is by itself a full-batch ERM problem which tends to be expensive to solve even approximately per-iteration. Plus, due to the potentially imbalanced data distribution over devices, the computational overload of the proximal point oracle could vary significantly across the network. *Therefore, it is important to investigate whether using minibatch stochastic approximation to the proximal point oracle (3) can provably improve the computational efficiency of `FedProx`.*

Last but not least, existing convergence analysis of `FedProx` mainly focuses on the empirical FL problem (2). The optimality in terms of the population FL problem (1) is not yet clear for `FedProx`. The primary goal of this work is to remedy these theoretical issues simultaneously, so as to lay a more solid theoretical foundation for the popularly applied `FedProx` algorithm.

## 1.2 Our Contributions

In this paper, we make progress towards understanding the convergence behavior of `FedProx` for non-convex heterogenous FL under weaker yet more realistic conditions. The main results are a set of local dissimilarity invariant bounds for smooth or non-smooth problems.

**Main results for the vanilla `FedProx`.** As a starting point to address the restrictiveness of local dissimilarity assumption, we provide a novel convergence analysis for the vanilla `FedProx` algorithm invariant to the $(B, 0)$-LGD condition. For smooth and non-convex optimization problems, our result in Theorem 1 shows that the rate of convergence to a stationary point is upper bounded by

$$\frac{1}{T}\sum_{t=0}^{T-1} \mathbb{E}\left[\left\|\nabla \bar{R}_{\text{erm}}(w_t)\right\|^2\right] \lesssim \max\left\{\frac{1}{T^{2/3}}, \frac{1}{\sqrt{TI}}\right\}, \tag{4}$$

where $I$ is the number devices randomly selected for local update at each iteration. If all the devices participate in the local updates for every round, i.e. $I_t = [M]$, the rate of convergence can be improved to $\mathcal{O}(\frac{1}{T^{2/3}})$. For $T < I^3$, the rate in (4) is dominated by $\mathcal{O}(\frac{1}{T^{2/3}})$ which gives the communication complexity $\frac{1}{\epsilon^{3/2}}$ to achieve an $\epsilon$-stationary solution. On the other hand when $T \ge I^3$, the rate is dominated by $\mathcal{O}(\frac{1}{\sqrt{TI}})$ which gives the communication complexity $\frac{1}{I\epsilon^2}$. Compared to the already known $\mathcal{O}(\frac{1}{\epsilon})$ complexity bound of `FedProx` under the unrealistic $(B, 0)$-LGD condition (Li et al., 2020b), our rate in (4) is slower but it holds without needing to impose stringent regularity conditions

on the dissimilarity of local functions, and it reveals the benefit of device minibatch sampling for accelerating convergence. Further for *non-smooth* and weakly convex problems, we establish in Theorem 2 the following rate of convergence regarding proper $\rho$-Moreau-envelopes of $\bar{R}_{\texttt{erm}}$:

$$\frac{1}{T} \sum_{t=0}^{T-1} \mathbb{E}\left[\left\|\nabla \bar{R}_{\texttt{erm},\rho}(w_t)\right\|^2\right] \lesssim \frac{1}{\sqrt{T}}, \tag{5}$$

The bound is not dependent on the number of selected devices in each round. In the case of $I = \mathcal{O}(1)$, the bounds in (4) and (5) are comparable, which indicates that smoothness is not must-have for `FedProx` to get sharper convergence bound especially with low participation ratio. On the other end when $I = \mathcal{O}(M)$, the bound (5) for non-smooth problems is slower than the bound (4) for smooth functions in large-scale networks.

**Main results for minibatch stochastic `FedProx`.** Then as the chief contribution of the present work, we propose a minibatch stochastic extension of `FedProx` along with its population optimization performance analysis from a novel perspective of algorithmic stability theory. Inspired by the recent success of minibatch stochastic proximal point methods (MSPP) (Li et al., 2014; Wang et al., 2017; Asi et al., 2020; Deng and Gao, 2021), we propose to implement `FedProx` using MSPP as the local update oracle. The resulting method, which is referred to as `FedMSPP`, is expected to attain improved trade-off between computation, communication and memory efficiency for large-scale FL. In the case of imbalanced data distribution, minibatching is also beneficial for making the local computation more balanced across the devices. Based on some extended uniform stability arguments for gradients, we show in Theorem 3 the following rate of convergence for `FedMSPP` in terms of population optimality, which is also invariant to the $(B, 0)$-LGD condition:

$$\frac{1}{T} \sum_{t=0}^{T-1} \mathbb{E}\left[\left\|\nabla \bar{R}(w_t)\right\|^2\right] \lesssim \max\left\{\frac{1}{T^{2/3}}, \frac{1}{\sqrt{TbI}}\right\}, \tag{6}$$

where $b$ is the minibatch size of local update. For empirical FL, identical bound holds under sampling according to empirical distribution. For $T < (bI)^3$, the rate in (6) is dominated by $\mathcal{O}(\frac{1}{T^{2/3}})$ which gives the communication complexity $\frac{1}{\epsilon^{3/2}}$, and it matches that of the vanilla `FedProx`. For sufficiently large $T \geq (bI)^3$, the rate is dominated by $\mathcal{O}(\frac{1}{\sqrt{TbI}})$ which gives the communication complexity $\frac{1}{bI\epsilon^2}$. This shows that local minibatching and device sampling are both beneficial for linearly speeding up communication. Further, when applied to non-smooth problems, we show in Theorem 4 that `FedMSPP` converges at the following rate with respect to proper $\rho$-Moreau-envelopes of $\bar{R}$:

$$\frac{1}{T} \sum_{t=0}^{T-1} \mathbb{E}\left[\left\|\nabla \bar{R}_\rho(w_t)\right\|^2\right] \lesssim \frac{1}{\sqrt{T}},$$

which is comparable to that of (6) when $b = \mathcal{O}(1)$ and $I = \mathcal{O}(1)$, but without witnessing the effect of linear speedup with respect to $b$ and $I$.

**Comparison with prior results.** In Table 1, we summarize our communication complexity bounds for `FedProx` (`FedMSPP`) and compare them with several related heterogeneous FL algorithms in terms of the conditions on local dissimilarity, applicability to non-smooth problems and tolerance to partial participation. A few observations are in order. *First*, regarding the local dissimilarity condition, all of our $\mathcal{O}(\frac{1}{\epsilon^2})$ bounds are not dependent on the $(B, H)$-LGD type conditions, and they are comparable to those of `SCAFFOLD` and `FCO` (for convex problems) which are also invariant to local dissimilarity conditions. *Second*, with regard to the applicability to non-smooth optimization, our convergence guarantees in Theorem 2 and Theorem 4 are established for non-smooth and weakly convex functions. While `FCO` is the only one in the other considered algorithms that can be applied to non-smooth problems, it is customized for federated convex composite optimization with potentially non-smooth regularizers (Yuan et al., 2021). *Third*, in terms of tolerance to partial participation, all of our results are robust to device sampling, and the $\mathcal{O}(\frac{1}{bI\epsilon^2})$ bound in Theorem 3 for `FedMSPP` is comparable to the best known results under partial participation as achieved by `FedAvg` and `SCAFFOLD`. If assuming that all the devices participate in local update for each communication round and using momentum acceleration techniques, substantially faster $\mathcal{O}(\frac{1}{\epsilon})$ bounds are possible for `STEM` and `FedPD`, while the $\mathcal{O}(\frac{1}{\epsilon^{3/2}})$ bounds can be achieved by `FedAvg` (Khanduri et al., 2021). To summarize the comparison, our $(B, H)$-LGD invariant convergence bounds for `FedProx` (`FedMSPP`) are comparable to the best-known rates in the identical setting, while covering the generic non-smooth and non-convex cases which to our knowledge has not been provably possible for other FL algorithms.

| Method | Work | Commun. Complex. | LD Condition | NS | PP |
|---|---|---|---|---|---|
| `FedProx` | (Li et al., 2020b) | $\mathcal{O}\left(\frac{1}{\epsilon}\right)$ | $(B,0)$-LGD | ✗ | ✓ |
| | Theorem 1 (ours) | $\mathcal{O}\left(\frac{1}{I\epsilon^2} + \frac{1}{\epsilon^{3/2}}\right)$ | – | ✗ | ✓ |
| | Theorem 2 (ours) | $\mathcal{O}\left(\frac{1}{\epsilon^2}\right)$ | – | ✓ | ✓ |
| `FedMSPP` | Theorem 3 (ours) | $\mathcal{O}\left(\frac{1}{bI\epsilon^2} + \frac{1}{\epsilon^{3/2}}\right)$ | – | ✗ | ✓ |
| | Theorem 4 (ours) | $\mathcal{O}\left(\frac{1}{\epsilon^2}\right)$ | – | ✓ | ✓ |
| `FedAvg` | (Karimireddy et al., 2020) | $\mathcal{O}\left(\frac{1}{bI\epsilon^2} + \frac{1}{\epsilon^{3/2}} + \frac{1}{\epsilon}\right)$ | $(B,H)$-LGD | ✗ | ✓ |
| | (Yu et al., 2019) | $\mathcal{O}\left(\frac{1}{bM\epsilon^2} + \frac{Mb}{\epsilon}\right)$ | $(0,H)$-LGD | ✗ | ✗ |
| | (Khanduri et al., 2021) | $\mathcal{O}\left(\frac{1}{\epsilon^{3/2}}\right)$ | – | ✗ | ✗ |
| `SCAFFOLD` | (Karimireddy et al., 2020) | $\mathcal{O}\left(\frac{1}{bI\epsilon^2} + \frac{(M/I)^{2/3}}{\epsilon}\right)$ | – | ✗ | ✓ |
| `FedPD` | (Zhang et al., 2020) | $\mathcal{O}\left(\frac{1}{\epsilon}\right)$ | – | ✗ | ✗ |
| `STEM` | (Khanduri et al., 2021) | $\mathcal{O}\left(\frac{1}{\epsilon}\right)$ | – | ✗ | ✗ |
| `FCO` | (Yuan et al., 2021) | $\mathcal{O}\left(\frac{1}{bM\epsilon^2} + \frac{1}{\epsilon}\right)$ (convex composite) | – | ✓ | ✗ |

Table 1: Comparison of heterogeneous FL algorithms in terms of communication complexity bounds for reaching an $\epsilon$-stationary solution, local dissimilarity (LD) condition, applicability to non-smooth (NS) functions and tolerance to partial participation (PP). Except for `FCO`, all the results listed are for non-convex functions. The involved quantities are $M$: total number of devices; $I$: number of chosen devices for partial participation; $b$: minibatch size for local stochastic optimization.

**Highlight of contributions.** The theoretical contributions of this work are highlighted as follows:

- From the perspective of algorithmic stability theory, we provide a set of novel local dissimilarity invariant convergence guarantees for the widely used `FedProx` algorithm for non-convex heterogeneous FL, with smooth or non-smooth local functions. Our theory for the first time reveals that local dissimilarity and smoothness are not necessary to guarantee the convergence of `FedProx` with reasonable rates.

- We present `FedMSPP` as a minibatch stochastic extension of `FedProx` and analyze its population optimization performance for both smooth and non-smooth FL problems, again without assuming local dissimilarity conditions. Particularly for smooth problems, our result provably shows that `FedMSPP` enjoys linear speedup in terms of minibatching size and partial participation ratio.

**Paper organization.** In Section 2 we present our local dissimilarity invariant convergence analysis for the vanilla `FedProx` with smooth or non-smooth loss functions. In Section 3 we propose `FedMSPP` as a minibatch stochastic extension of `FedProx` and analyze its convergence behavior through the lens of algorithmic stability theory. In Section 4, we discuss some additional related work on the topics covered by this paper. The concluding remarks are made in Section 5. Finally, all the technical proofs and some additional related work are relegated to the appendix sections.

## 2 Convergence of `FedProx`

We begin by providing an improved analysis for the vanilla `FedProx` which is not relying on the $(B, H)$-LGD type conditions. We first introduce notations that will be used in the analysis to follow.

**Notations.** Throughout the paper, we use $[n]$ to denote the set $\{1, ..., n\}$, $\|\cdot\|$ to denote the Euclidean norm and $\langle\cdot, \cdot\rangle$ to denote the Euclidean inner product. We say a function $f$ is $G$-Lipschitz continuous if $|f(w) - f(w')| \leq G\|w - w'\|$ for all $w, w' \in \mathbb{R}^p$, and it is $L$-smooth if $|\nabla f(w) - \nabla f(w')| \leq L\|w - w'\|$ for all $w, w' \in \mathbb{R}^p$. Moreover, we say $f$ is $\nu$-weakly convex if for any $w, w' \in \mathbb{R}^p$,

$$f(w) \geq f(w') + \langle \partial f(w'), w - w' \rangle - \frac{\nu}{2}\|w - w'\|^2,$$

where $\partial f(w')$ represents a subgradient of $f$ evaluated at $w'$. We denote by

$$f_\eta(w) := \min_u \left\{ f(u) + \frac{1}{2\eta}\|u - w\|^2 \right\}$$

the $\eta$-Moreau-envelope of $f$, and by

$$\mathrm{prox}_{\eta f}(w) := \arg\min_u \left\{ f(u) + \frac{1}{2\eta} \|u - w\|^2 \right\}$$

the proximal mapping associated with $f$. We also need to access the following definition of inexact local update oracle for FedProx.

**Definition 2** (Local inexact oracle of FedProx). *Suppose that the local proximal point regularized objective $Q_{erm}^{(m)}(w; w_{t-1})$ (cf. (3)) admits a global minimizer. For each time instance $t$, we say that the local update oracle of FedProx is $\varepsilon_t$-inexactly solved with sub-optimality $\varepsilon_t \geq 0$ if*

$$Q_{erm}^{(m)}(w_t^{(m)}; w_{t-1}) \leq \min_w Q_{erm}^{(m)}(w; w_{t-1}) + \varepsilon_t.$$

Throughout our analysis, we focus on the case where the devices are sampled with replacement, while all the results extend well to the regime of sampling without replacement.

## 2.1 Results for Smooth Problems

The following theorem is our main result on the convergence rate of FedProx for smooth and non-convex federated optimization problems. A proof of this result is deferred to Appendix B.1. We assume that the initial sub-optimality $\bar{\Delta}_{erm}^{(0)} := \bar{R}_{erm}(w_0) - \min_{w \in \mathbb{R}^p} \bar{R}_{erm}(w)$ is bounded.

**Theorem 1.** *Assume that for each $m \in [M]$, the loss function $\ell^{(m)}$ is $G$-Lipschitz and $L$-smooth with respect to its first argument. Set $|I_t| \equiv I$ and $\eta_t \equiv \frac{1}{3L} \min\left\{ \frac{1}{T^{1/3}}, \sqrt{\frac{I}{T}} \right\}$. Suppose that the local update oracle of FedProx is $\varepsilon_t$-inexactly solved with $\varepsilon_t \leq \min\left\{ \frac{2L^2 G^2 \eta_t^3}{I^2 (L\eta_t+1)}, \frac{G^2 \eta_t}{2I(L\eta_t+1)} \right\}$. Let $t^*$ be an index uniformly randomly chosen in $\{0, 1, ..., T-1\}$. Then it holds that*

$$\mathbb{E}\left[ \left\| \nabla \bar{R}_{erm}(w_{t^*}) \right\|^2 \right] \lesssim \left( L\bar{\Delta}_{erm}^{(0)} + G^2 \right) \max\left\{ \frac{1}{T^{2/3}}, \frac{1}{\sqrt{TI}} \right\}.$$

A few remarks are in order.

**Remark 1.** *Compared to the $\mathcal{O}\left(\frac{1}{T}\right)$ bound from Li et al. (2020b), our rate established in Theorem 1 is slower but it is valid without assuming the unrealistic $(B, 0)$-LGD conditions and imposing strong regularization conditions on $I$ (see, e.g., Li et al., 2020b, Remark 5). Moreover, the dominant term $\frac{1}{\sqrt{TI}}$ in our bound reveals the benefit of device sampling for linear speedup which is not clear in the previous analysis by Li et al. (2020b).*

**Remark 2.** *In the extreme case of full device participation, i.e., $I_t \equiv [M]$, the terms related to $I$ in Theorem 1 can be removed and thus the convergence rate becomes $\frac{1}{T^{2/3}}$ under $\eta_t = \mathcal{O}\left(\frac{1}{LT^{1/3}}\right)$. In this same setting, we comment that the rate can also be improved to $\mathcal{O}\left(\frac{1}{T}\right)$ using our proof augments if $(B, 0)$-LGD is additionally assumed.*

**Remark 3.** *The $G$-Lipschitz-loss assumption in Theorem 1 can be alternatively replaced by the bounded gradient condition as commonly used in the analysis of FL algorithms (Li et al., 2020b; Zhang et al., 2020). Despite that our analysis does not rely on the $(B, 0)$-LGD condition, the assumed $G$-Lipschitz (or bounded gradient) condition actually implies that the local objective gradients are not too dissimilar, which shares a close spirit to the typically assumed $(0, H)$-LGD condition (Karimireddy et al., 2020) and inter-client-variance condition (Khanduri et al., 2021). It is noteworthy that these mentioned client heterogeneity conditions are substantially milder than the $(B, 0)$-LGD condition as required in the original analysis of FedProx.*

## 2.2 Results for Non-smooth Problems

Now we turn to study the convergence of FedProx for weakly convex but not necessarily smooth problems. For the sake of presentation clarity, we work on the exact FedProx in which the local update oracle is assumed to be exactly solved, i.e. $\varepsilon_t \equiv 0$. Extension to the inexact case is more or less straightforward, though with somewhat more involved perturbation treatments. In the analysis to follow, we assume that the initial sub-optimality $\bar{\Delta}_{erm,\rho}^{(0)} := \bar{R}_{erm,\rho}(w_0) - \min_w \bar{R}_{erm,\rho}(w)$ associated with $\rho$-Moreau-envelope of $\bar{R}_{erm}$ is bounded. The following is our main result on the convergence of FedProx for non-smooth and weakly convex problems.

**Theorem 2.** *Assume that for each $m \in [M]$, the loss function $\ell^{(m)}$ is $G$-Lipschitz and $\nu$-weakly convex with respect to its first argument. Set $\eta_t \equiv \frac{\rho}{\sqrt{T}}$ for arbitrary $\rho < \frac{1}{2\nu}$. Suppose that the local update oracle of* `FedProx` *is exactly solved with $\varepsilon_t \equiv 0$. Let $t^*$ be an index uniformly randomly chosen in $\{0, 1, ..., T-1\}$. Then it holds that*

$$\mathbb{E}\left[\left\|\nabla \bar{R}_{erm,\rho}(w_{t^*})\right\|^2\right] \lesssim \frac{\bar{\Delta}_{erm,\rho}^{(0)} + \rho G^2}{\rho \sqrt{T}}.$$

*Proof.* The proof technique is inspired by the arguments from Davis and Drusvyatskiy (2019) developed for analyzing stochastic model-based algorithms, with several new elements along developed for handling the challenges introduced by the model averaging and partial participation mechanisms associated with `FedProx`. A particular crux here is that due to the random subset model aggregation of $w_t = \frac{1}{|I_t|} \sum_{\xi \in I_t} w_t^{(\xi)}$, the local function values $R_{\text{erm}}^{(\xi)}(w_t)$ are no longer independent of each other though $\xi$ is uniformly random. As a consequence, $\frac{1}{|I_t|} \sum_{\xi \in I_t} R_{\text{erm}}^{(\xi)}(w_t)$ is *not* an unbiased estimation of $\bar{R}_{\text{erm}}(w_t)$. To overcome this technical obstacle, we make use of a key observation that $w_t^{(m)}$ will be almost surely close enough to $w_{t-1}$ if the learning rate $\eta_t$ is small enough (which is the case in our choice of $\eta_t$), and thus we can replace the former with the latter whenever beneficial but without introducing too much approximation error. See Appendix B.2 for a full proof of this result. $\square$

A few remarks are in order.

**Remark 4.** *To our best knowledge, Theorem 2 is the first convergence guarantee for FL algorithms applicable to generic non-smooth and weakly convex problems. This is in sharp contrast with* `FCO` *(Yuan et al., 2021) which focuses on composite convex and non-smooth problems such as $\ell_1$-estimation, or* `Fed-HT` *(Tong et al., 2020) which is specially customized for cardinality-constrained sparse learning problems where the non-convexity essentially arises from the cardinality constraint.*

**Remark 5.** *Let us consider $\bar{w}_{t^*} := prox_{\rho \bar{R}_{erm}}(w_{t^*})$, the proximal mapping of $w_{t^*}$ associated with $\bar{R}_{erm}$. In view of a feature of Moreau envelope to characterize stationarity (Davis and Drusvyatskiy, 2019), if $w_{t^*}$ has small gradient norm $\left\|\nabla \bar{R}_{erm,\rho}(w_{t^*})\right\|$, then $\bar{w}_{t^*}$ must be a near-stationary solution and $w_{t^*}$ stays in the proximity of $\bar{w}_{t^*}$ due to the identity $\|w_{t^*} - \bar{w}_{t^*}\| = \rho \left\|\nabla \bar{R}_{erm,\rho}(w_{t^*})\right\|$. Therefore, the bound in Theorem 2 suggests that in expectation $\bar{w}_{t^*}$ converges to a stationary solution and $w_{t^*}$ converges to $\bar{w}_{t^*}$, both at the rate of $\mathcal{O}\left(\frac{1}{\sqrt{T}}\right)$.*

**Remark 6.** *We comment that the bound in Theorem 2 is not dependent on $I$, the number of selected devices. On one hand, for $I = \mathcal{O}(1)$ and sufficiently large $T > \mathcal{O}(I^3)$, the bounds Theorem 1 and Theorem 2 are comparable to each other, which demonstrates that the smoothness is not must-have for* `FedProx` *to get sharper convergence bound with small device sampling rate. On the other hand, in the near-full participation setting where $I = \mathcal{O}(M)$, the bound in Theorem 2 for non-smooth problems will be slower when $M$ is large. Extremely when $I_t = [M]$, the $\mathcal{O}\left(\frac{1}{\sqrt{T}}\right)$ bound is substantially inferior to the smooth case which has improved rate of $\mathcal{O}\left(\frac{1}{T^{2/3}}\right)$ as discussed in Remark 2.*

## 3 Convergence of `FedProx` with Stochastic Minibatching

When it comes to the implementation of `FedProx`, a notable challenge is that the local proximal point update oracle (3) is by itself a full-batch ERM problem which would be expensive to solve even approximately in large-scale settings. Moreover, in the settings where the data distribution over devices is highly imbalanced, the computational overload of local update could vary significantly across the network, which impairs communication efficiency. It is thus desirable to seek stochastic approximation schemes for hopefully improving the local oracle update efficiency and overload balance of `FedProx`. To this end, inspired by the recent success of minibatch stochastic proximal point methods (MSPP) (Asi et al., 2020; Deng and Gao, 2021), we propose to implement `FedProx` using MSPP as the local stochastic optimization oracle. More precisely, let $B_t^{(m)} = \{z_{i,t}^{(m)}\}_{i=1}^b \overset{\text{i.i.d.}}{\sim} (\mathcal{D}^{(m)})^b$ be a minibatch of $b$ i.i.d. samples drawn from the distribution $\mathcal{D}^{(m)}$ at device $m$ and time instance $t \geq 1$. We denote

$$R_{B_t^{(m)}}^{(m)}(w) := \frac{1}{b} \sum_{i=1}^b \ell^{(m)}(w; z_{i,t}^{(m)}) \tag{7}$$

---

**Algorithm 1:** FedMSPP: Federated Minibatch Stochastic Proximal Point

---

**Input** : Minibatch size $b$; learning rates $\{\gamma_t\}_{t \in [T]}$.
**Output** : $w_T$.
**Initialization** Set $w_0$, *e.g., typically as a zero vector.*
**for** $t = 1, 2, ..., T$ **do**

    | /* Device selection and model broadcast on the server              */
    | Server uniformly randomly selects a subset $I_t \subseteq [M]$ of devices and sends $w_{t-1}$ to all the
    | selected devices;
    | /* Local model updates on the selected devices                   */
    | **for** $\xi \in I_t$ *in parallel* **do**

        | Device $\xi$ samples a minibatch $B_t^{(\xi)} = \{z_{i,t}^{(\xi)}\}_{i=1}^b \overset{\text{i.i.d.}}{\sim} (\mathcal{D}^{(\xi)})^b$.
        | Device $\xi$ inexactly updates the its local model as

$$w_t^{(\xi)} \approx \underset{w \in \mathbb{R}^p}{\arg\min} \left\{ Q_{B_t^{(\xi)}}^{(\xi)}(w; w_{t-1}) := R_{B_t^{(\xi)}}^{(\xi)}(w) + \frac{1}{2\eta_t} \|w - w_{t-1}\|^2 \right\}, \qquad (8)$$

        | where $R_{B_t^{(\xi)}}^{(\xi)}(w)$ is given by (7).
        | Device $\xi$ sends $w_t^{(\xi)}$ back to server.

    | **end**
    | /* Model aggregation on the server                            */
    | Sever aggregates the local models received from $I_t$ to update the global model as
    | $w_t = \frac{1}{|I_t|} \sum_{\xi \in I_t} w_t^{(\xi)}$.

**end**

---

as the local minibatch empirical risk function over $B_t^{(m)}$. Here, the only modification we propose to make is to replace the empirical risk $R_{\text{erm}}^{(m)}(w)$ in the original update form (3) with its minibatch counterpart $R_{B_t^{(m)}}^{(m)}(w)$. The resultant FL framework, which we refer to as `FedMSPP` (Federated MSPP), is outlined in Algorithm 1. Clearly, the vanilla `FedProx` is a special case of `FedMSPP` when applied to the federated ERM form (2) with full data batch $B_t^{(m)} \equiv D^{(m)}$.

### 3.1 Results for Smooth Problems

We first analyze the convergence rate of `FedMSPP` for smooth and non-convex problems using the tools borrowed from algorithmic stability theory. Analogous to the Definition 2, we introduce the following definition of inexact local update oracle for `FedMSPP`.

**Definition 3** (Local inexact oracle of `FedMSPP`). *Suppose that the local proximal point regularized objective $Q_{B_t^{(m)}}^{(m)}(w; w_{t-1})$ (cf. (8)) admits a global minimizer. For each time instance $t$, we say that the local update oracle of* **FedMSPP** *is $\varepsilon_t$-inexactly solved with sub-optimality $\varepsilon_t \geq 0$ if*

$$Q_{B_t^{(m)}}^{(m)}(w_t^{(m)}; w_{t-1}) \leq \min_w Q_{B_t^{(m)}}^{(m)}(w; w_{t-1}) + \varepsilon_t.$$

We also assume that the initial population sub-optimality $\bar{\Delta}^{(0)} = \bar{R}(w^{(0)}) - \min_{w \in \mathbb{R}^p} \bar{R}(w)$ is bounded. The following theorem is our main result on `FedMSPP` for smooth and non-convex FL problems.

**Theorem 3.** *Assume that for each $m \in [M]$, the loss function $\ell^{(m)}$ is $G$-Lipschitz and $L$-smooth with respect to its first argument. Set $|I_t| \equiv I$ and $\eta_t \equiv \frac{1}{8L} \min \left\{ \frac{1}{T^{1/3}}, \sqrt{\frac{bI}{T}} \right\}$. Suppose that the local update oracle of* **FedMSPP** *is $\varepsilon_t$-inexactly solved with $\varepsilon_t \leq \min \left\{ \frac{G^2 \eta_t}{2(L\eta_t+1)}, \frac{G^2 \eta_t}{8b^2}, \frac{L^2 G^2 \eta_t^3}{2bI(L\eta_t+1)} \right\}$. Let $t^*$ be an index uniformly randomly chosen in $\{0, 1, ..., T-1\}$. Then it holds that*

$$\mathbb{E}\left[ \|\nabla \bar{R}(w_{t^*})\|^2 \right] \lesssim \left( L\bar{\Delta}^{(0)} + G^2 \right) \max \left\{ \frac{1}{T^{2/3}}, \frac{1}{\sqrt{TbI}} \right\}.$$

*Proof.* Let us consider $d_t^{(m)} = \nabla R_{B_t^{(m)}}^{(m)}(w_t^{(m)})$ which is roughly the local update direction on device $m$, in the sense that $w_t^{(m)} \approx w_{t-1} - \eta_t d_t^{(m)}$ given that the local update oracle is solved to sufficient accuracy. As a key ingredient of our proof, we show via some extended uniform stability arguments in terms of gradients (see Lemma 3) that the averaged directions $d_t := \frac{1}{|I_t|} \sum_{\xi \in I_t} d_t^{(\xi)}$ aligns well with the global gradient $\nabla \bar{R}(w_{t-1})$ in expectation (see Lemma 11). Therefore, in average it roughly holds that $w_t = \frac{1}{|I_t|} \sum_{\xi \in I_t} w_t^{(\xi)} \approx w_{t-1} - \eta_t d_t \approx w_{t-1} - \eta_t \nabla \bar{R}(w_{t-1})$, which suggests that $w_t$ is updated roughly along the direction of global gradient descent and thus is expected to converge quickly. Based on this novel analysis, we are free of explicitly imposing local dissimilarity type conditions on local objectives. See Appendix C.1 for a full proof of this result. □

**Remark 7.** *For $T \geq (bI)^3$, the bound in Theorem 3 is dominated by $\mathcal{O}\left(\frac{1}{\sqrt{TbI}}\right)$ which gives the communication complexity $\frac{1}{bI\epsilon^2}$. This shows that* `FedMSPP` *enjoys linear speedup with respect to both local minibatching and device sampling sizes.*

**Remark 8.** *While the bound in Theorem 3 is derived for the population form of FL in (1), an identical bound naturally holds for the empirical form (2) under minibatch sampling according to local data empirical distribution.*

## 3.2 Results for Non-smooth Problems

Analogues to `FedProx` , we can further show that `FedMSPP` converges reasonably well when applied to weakly convex and non-smooth problems. In the analysis to follow, we assume that the initial sub-optimality $\bar{\Delta}_\rho^{(0)} := \bar{R}_\rho(w_0) - \min_{w \in \mathbb{R}^p} \bar{R}_\rho(w)$ associated with $\rho$-Moreau-envelope of $\bar{R}$ is bounded. The following is our main result in this line.

**Theorem 4.** *Assume that for each $m \in [M]$, the loss function $\ell^{(m)}$ is $G$-Lipschitz and $\nu$-weakly convex with respect to its first argument. Set $\eta_t \equiv \frac{\rho}{\sqrt{T}}$ for arbitrary $\rho < \frac{1}{2\nu}$. Suppose that the local update oracle of* `FedMSPP` *is exactly solved with $\varepsilon_t \equiv 0$. Let $t^*$ be an index uniformly randomly chosen in $\{0, 1, ..., T-1\}$. Then it holds that*

$$\mathbb{E}\left[\left\|\nabla \bar{R}_\rho(w_{t^*})\right\|^2\right] \lesssim \frac{\bar{\Delta}_\rho^{(0)} + \rho G^2}{\rho\sqrt{T}}.$$

*Proof.* The proof argument is a slight adaptation of that of Theorem 2 to the population FL setup (1) with `FedMSPP`. For the sake of completeness, a full proof is reproduced in Appendix C.2. □

We comment in passing that the discussions made in Remarks 4-6 extend directly to Theorem 4.

## 4 Additional Related Work

The present work is situated at the intersection of federated learning, stochastic proximal point optimization and algorithmic stability theory. We next briefly review some additional work in these lines of research that are closely related to ours.

*Heterogenous federated learning.* The presence of device heterogeneity features a key distinction between FL and classic distributed learning. The most commonly used FL method is `FedAvg` (McMahan et al., 2017), where the local update oracle is formed as multi-epoch SGD. `FedAvg` was early analyzed for identical functions (Stich, 2019; Stich and Karimireddy, 2020) under the name of local SGD. In heterogeneous setting, numerous recent studies have focused on the analysis of `FedAvg` and other variants under various notions of local dissimilarity (Li et al., 2020c; Woodworth et al., 2020; Chen et al., 2020; Khaled et al., 2020; Reddi et al., 2021; Khanduri et al., 2021; Li et al., 2022; Chen et al., 2022; Zhao et al., 2022). As another representative FL method, `FedProx` (Li et al., 2020b) has recently been proposed to apply averaged proximal point updates to solve heterogeneous federated minimization problems. The theoretical guarantees of `FedProx` have been established for both convex and non-convex problems, but under a fairly stringent assumption of gradient similarity (see Definition 1) to measure data heterogeneity (Li et al., 2020b; Pathak and Wainwright, 2020; Nguyen et al., 2021). This assumption was relaxed by `FedPD` (Zhang et al., 2020) inside a meta-framework of primal-dual optimization. The `SCAFFOLD` (Karimireddy et al., 2020) and

`VRL-SGD` (Liang et al., 2019) are two algorithms that utilize variance reduction techniques to correct the local update directions, achieving convergence guarantees independent of the data heterogeneity. For composite non-smooth FL problems, the `FCO` proposed in Yuan et al. (2021) employs a server dual averaging procedure to circumvent the curse of primal averaging suffered by `FedAvg`. In sharp contrast to these prior works which either require certain stringent local dissimilarity conditions, or require full device participation, or only applicable to smooth problems, we show through a novel analysis based on algorithmic stability theory that the well-known `FedProx` can elegantly overcome all these shortcomings in a simple algorithmic framework. Another research direction in FL is to adopt compression methods for efficient communication (e.g., Haddadpour et al. (2020); Li and Li (2022)), which could also be applied to `FedProx`, as a topic for future investigation.

*Minibatch stochastic proximal point methods.* The proposed `FedMSPP` algorithm is a variant of `FedProx` that simply replaces the local proximal point oracle with MSPP, which in each iteration updates the local model via (approximately) solving a proximal point estimator over a stochastic minibatch. The MSPP-type methods have been shown to attain a substantially improved iteration stability and adaptivity for large-scale machine learning, especially in non-smooth optimization settings (Li et al., 2014; Wang et al., 2017; Asi and Duchi, 2019; Deng and Gao, 2021). However, it is not yet known if `FedProx` or `FedMSPP` can achieve similar strong guarantees for non-smooth heterogenous FL problems.

*Algorithmic stability.* Our analysis for `FedMSPP` builds largely upon the classic algorithmic stability theory. Since the seminal work of Bousquet and Elisseeff (2002), algorithmic stability has been serving as a powerful proxy for establishing strong generalization bounds (Zhang, 2003; Mukherjee et al., 2006; Shalev-Shwartz et al., 2010). Particularly, the state-of-the-art risk bounds of strongly convex ERM are offered by approaches based on the notion of uniform stability (Feldman and Vondrák, 2018, 2019; Bousquet et al., 2020; Klochkov and Zhivotovskiy, 2021). It was shown by Hardt et al. (2016) that the solution obtained via SGD is stable for smooth convex or non-convex loss functions. For non-smooth convex losses, the stability induced generalization bounds have been established for SGD (Lei and Ying, 2020; Bassily et al., 2020; Yuan and Li, 2022). Through the lens of uniform algorithmic stability, convergence rates of MSPP have been studied for non-smooth and convex (Wang et al., 2017), or weakly convex (Deng and Gao, 2021) losses.

## 5 Conclusions

In this paper, we have exposed three shortcomings of the prior analysis for `FedProx` in unrealistic assumptions about local dissimilarity, inapplicability to non-smooth problems and expensive (and potentially imbalanced) computational cost of local update. In order to tackle these issues, we developed a novel convergence theory for the vanilla `FedProx` and its minibatch stochastic variant, `FedMSPP`, through the lens of algorithmic stability theory. In a nutshell, our results reveal that with minimal modifications, `FedProx` is able to kill three birds with one stone: it enjoys favorable rates of convergence which are simultaneously invariant to certain stringent local dissimilarity conditions, applicable to smooth or non-smooth problems, and scaling linearly with respect to local minibatch size and device sampling ratio for smooth problems. To the best of our knowledge, the present work is the first theoretical contribution that achieves all these appealing properties in a single FL framework.

**Limitations.** While our results in Theorems 2 and 4 for the first time guarantee the non-asymptotic convergence of `FedProx` and `FedMSPP` for non-smooth and weakly-convex problems, the corresponding rates of convergence so far cannot demonstrate any linear speedup effort w.r.t. device sampling ratio and local minibatching size. This is as opposed to what have been shown for smooth problems in Theorems 1 and 3; thus we view it as a potential limitation of the techniques used by our analysis. In the smooth-loss case, the comparison in Table 1 suggests that our results in Theorem 1 and Theorem 3 show no faster convergence rates than those of the existing FL methods based on local SGD update, despite that `FedProx`/`FedMSPP` requires a considerably more expensive oracle for local update.

## Acknowledgments and Disclosure of Funding

The authors sincerely thank the anonymous reviewers for their insightful comments which are truly helpful for improving this paper. The research was conducted while both authors worked for the Baidu Cognitive Computing Lab. Xiao-Tong Yuan is also funded in part by the National Key Research and Development Program of China under Grant No. 2018AAA0100400, and in part by the Natural Science Foundation of China (NSFC) under Grant No.U21B2049, No.61876090 and No.61936005.

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
