## A  Preliminaries

We present in this section some preliminary results on the classic algorithmic stability theory to be used in our analysis. Let us consider an algorithm $A : \mathcal{Z}^N \mapsto \mathcal{W}$ that maps a training data set $S = \{z_i\}_{i \in [N]} \in \mathcal{Z}^N$ to a model $A(S)$ in a closed subset $\mathcal{W} \subseteq \mathbb{R}^p$ such that the following population risk function (with a slight abuse of notation) evaluated at the model is as small as possible:

$$R(A(S)) := \mathbb{E}_{Z \sim \mathcal{D}}[\ell(A(S); Z)].$$

The corresponding empirical risk is defined by

$$R_S(A(S)) := \mathbb{E}_{Z \sim \texttt{Unif}(S)}[\ell(A(S); Z)] = \frac{1}{N} \sum_{i=1}^{N} \ell(A(S); z_i).$$

We denote by $S \doteq S'$ if a pair of data sets $S$ and $S'$ differ in a single data point. The following concept of stability that serves as a powerful tool for analyzing the generalization bounds of learning algorithms (Hardt et al., 2016; Elisseeff et al., 2005; Bassily et al., 2020).

**Definition 4** (Uniform Argument Stability). *Let $A : \mathcal{Z}^N \mapsto \mathcal{W}$ be a learning algorithm that maps a data set $S \in \mathcal{Z}^N$ to a model $A(S) \in \mathcal{W}$. Then $A$ is said to have $\gamma$-uniform stability if for every $N \geq 1$,*

$$\sup_{S \doteq S'} \|A(S) - A(S')\| \leq \gamma.$$

The following basic lemma is about the uniform argument stability of an inexact regularized empirical risk minimization (ERM) estimator. See Appendix D.1 for its proof.

**Lemma 1.** *Assume that the loss function $\ell$ is $G$-Lipschitz with respect to its first argument. Suppose that the regularized objective $R_S^r(w) := \frac{1}{N} \sum_{i=1}^{N} \ell(w; z_i) + r(w)$ is $\lambda$-strongly convex for any $S$. Consider the inexact estimator $w_S$ that satisfies the following for some $\varepsilon_t \geq 0$:*

$$R_S^r(w_S) \leq \min_w R_S^r(w) + \varepsilon_t.$$

*Then $w_S$ has uniform argument stability with parameter $\frac{4G}{\lambda N} + 2\sqrt{\frac{2\varepsilon_t}{\lambda}}$.*

We further need to use the following variant of Efron-Stein inequality to random vector-valued functions (see, e.g., Lemma 6, Rivasplata et al., 2018).

**Lemma 2** (Efron-Stein inequality for vector-valued functions). *Let $S = \{Z_1, Z_2, ..., Z_N\}$ be a set of i.i.d. random variables valued in $\mathcal{Z}$. Suppose that the function $h : \mathcal{Z}^N \mapsto \mathcal{H}$ valued in a Hilbert space $\mathcal{H}$ is measurable and satisfies the bounded differences property, i.e., the following inequality holds for any $i \in [N]$ and any $z_1, ..., z_N, z_i'$:*

$$\|h(z_1, ..., z_{i-1}, z_i, z_{i+1}, ..., z_N) - h(z_1, ..., z_{i-1}, z_i', z_{i+1}, ..., z_N)\| \leq \beta.$$

*Then it holds that*

$$\mathbb{E}_S \left[ \|h(S) - \mathbb{E}_S[h(S)]\|^2 \right] \leq \beta^2 N.$$

Based on the Efron-Stein inequality in Lemma 2, we can establish the following lemma which states the generalization bounds of a uniformly stable learning algorithm in terms of gradient. A proof of this result can be found in Appendix D.2.

**Lemma 3.** *Suppose that a learning algorithm $A : \mathcal{Z}^N \mapsto \mathcal{W}$ has $\gamma$-uniform stability. Assume that the loss function $\ell$ is $G$-Lipschitz and $L$-smooth with respect to its first argument. Then the following bounds hold:*

$$\|\mathbb{E}_S [\nabla R(A(S)) - \nabla R_S(A(S))]\| \leq L\gamma,$$

$$\mathbb{E}_S \left[ \|\nabla R(A(S)) - \mathbb{E}_S [\nabla R(A(S))]\|^2 \right] \leq L^2 \gamma^2 N.$$

# B Proofs for Section 2

## B.1 Proof of Theorem 1

Let $d_t^{(m)} = \nabla R_{\mathtt{erm}}^{(m)}(w_t^{(m)})$. We define the following quantities

$$d_t := \frac{1}{|I_t|} \sum_{\xi \in I_t} d_t^{(\xi)}, \quad \bar{d}_t := \frac{1}{M} \sum_{m=1}^{M} d_t^{(m)}. \tag{9}$$

The following elementary lemma is useful in our analysis.

**Lemma 4.** *Assume that for each* $m \in [M]$, *the loss function* $\ell^{(m)}$ *is G-Lipschitz. Set* $|I_t| \equiv I$. *Then it holds that*

$$\mathbb{E}[d_t] = \bar{d}_t, \quad \mathbb{E}[\|d_t - \bar{d}_t\|^2] \leq \frac{G^2}{I}.$$

*Proof.* By uniform sampling strategy we have

$$\mathbb{E}[d_{I_t}] = \mathbb{E}\left[\frac{1}{|I_t|} \sum_{\xi \in I_t} d_t^{(\xi)}\right] = \frac{1}{I} \sum_{\xi \in I_t} \mathbb{E}\left[d_t^{(\xi)}\right] = \frac{1}{I} \sum_{\xi \in I_t} \frac{1}{M} \sum_{m=1}^{M} d_t^{(m)} = \bar{d}_t.$$

Then it follows that

$$
\begin{aligned}
\mathbb{E}[\|d_t - \bar{d}_t\|^2] &= \mathbb{E}\left[\left\|\frac{1}{|I_t|} \sum_{\xi \in I_t} d_t^{(\xi)} - \bar{d}_t\right\|^2\right] \\
&= \frac{1}{I^2} \mathbb{E}\left[\left\|\sum_{\xi \in I_t} (d_t^{(\xi)} - \bar{d}_t)\right\|^2\right] \\
&= \frac{1}{I^2} \sum_{\xi \in I_t} \mathbb{E}\left[\left\|d_t^{(\xi)} - \bar{d}_t\right\|^2\right] \leq \frac{1}{I} \mathbb{E}\left[(d_t^{(\xi)})^2\right] \leq \frac{G^2}{I},
\end{aligned}
$$

where we have used the fact $\mathbb{E}\left[d_t^{(\xi)}\right] = \bar{d}_t$, the independence among the indices in $I_t$ and the $G$-Lipschitzness of losses. The desired bounds are proved. $\qquad\square$

We also need the following lemma which quantifies the impact of local update precision to the gradient norm at the inexact solution.

**Lemma 5.** *Assume that for each* $m \in [M]$, *the loss function* $\ell^{(m)}$ *is L-smooth with respect to its first argument. Suppose that the local update oracle of* `FedProx` *is* $\varepsilon_t$-*inexactly solved and* $\eta_t < \frac{1}{L}$. *Then it holds that*

$$\left\|w_t^{(m)} - w_{t-1} + \eta_t d_t^{(m)}\right\| \leq \eta_t \sqrt{2(L + \eta_t^{-1})\varepsilon_t}.$$

*Proof.* Recall $Q_{\mathtt{erm}}^{(m)}(w; w_{t-1}) = R_{\mathtt{erm}}^{(m)}(w) + \frac{1}{2\eta_t}\|w - w_{t-1}\|^2$. Since the loss functions are $L$-smooth and $\eta_t < \frac{1}{L}$, $Q_{\mathtt{erm}}^{(m)}(w; w_{t-1})$ is strongly convex and thus admits a global minimizer. Then we have

$$
\begin{aligned}
&\left\|\nabla R_{\mathtt{erm}}^{(m)}(w_t^{(m)}) + \frac{1}{\eta_t}(w_t^{(m)} - w_{t-1})\right\|^2 \\
&= \left\|\nabla Q_{\mathtt{erm}}^{(m)}(w_t^{(m)}; w_{t-1})\right\|^2 \\
&\leq 2(L + \eta_t^{-1})\left(Q_{\mathtt{erm}}^{(m)}(w_t^{(m)}; w_{t-1}) - \min_w Q_{\mathtt{erm}}^{(m)}(w; w_{t-1})\right) \leq 2(L + \eta_t^{-1})\varepsilon_t,
\end{aligned}
$$

where in the last inequality is due to Definition 2. This implies the desired bound. $\qquad\square$

**Lemma 6.** *Assume that for each $m \in [M]$, the loss function $\ell^{(m)}$ is G-Lipschitz and L-smooth with respect to its first argument. Suppose that the local update oracle of* `FedProx` *is $\varepsilon_t$-inexactly solved and $\eta_t < \frac{1}{L}$. Then the following holds almost surely:*

$$\|\nabla \bar{R}_{erm}(w_{t-1}) - \bar{d}_t\|^2 \leq L^2 \left( G + \sqrt{2(L + \eta_t^{-1})\varepsilon_t} \right)^2 \eta_t^2.$$

*Proof.* By Lemma 5 we know that

$$\|w_t^{(m)} - w_{t-1}\| \leq \eta_t \|d_t^{(m)}\| + \eta_t \sqrt{2(L + \eta_t^{-1})\varepsilon_t} \leq \left( G + \sqrt{2(L + \eta_t^{-1})\varepsilon_t} \right) \eta_t, \qquad (10)$$

where we have used the $G$-Lipschitz assumption of loss. By definition we can see that

$$
\begin{aligned}
\|\nabla \bar{R}_{\mathtt{erm}}(w_{t-1}) - \bar{d}_t\|^2 &= \left\| \frac{1}{M} \sum_{m=1}^{M} \left( \nabla R_{\mathtt{erm}}^{(m)}(w_{t-1}) - \nabla R_{\mathtt{erm}}^{(m)}(w_t^{(m)}) \right) \right\|^2 \\
&\leq \frac{1}{M} \sum_{m=1}^{M} \left\| \nabla R_{\mathtt{erm}}^{(m)}(w_{t-1}) - \nabla R_{\mathtt{erm}}^{(m)}(w_t^{(m)}) \right\|^2 \\
&\overset{\zeta_1}{\leq} \frac{L^2}{M} \sum_{m=1}^{M} \left\| w_{t-1} - w_t^{(m)} \right\|^2 \\
&\overset{\zeta_2}{\leq} L^2 \left( G + \sqrt{2(L + \eta_t^{-1})\varepsilon_t} \right)^2 \eta_t^2,
\end{aligned}
$$

where in "$\zeta_1$" we have used the $L$-smoothness of loss, in "$\zeta_2$" we have used (10). This proves the desired bound. $\qquad\square$

With all the above lemmas in place, we can prove the main result in Theorem 1. Let $\{\mathcal{F}_t\}_{t \geq 1}$ be the filtration generated by the random iterates $\{w_t\}_{t \geq 1}$ as $\mathcal{F}_t = \sigma(w_1, w_2, ..., w_t)$, where the randomness comes from the sampling of devices for partial participation.

*Proof of Theorem 1.* Let us denote $\delta_t^{(m)} := \eta_t^{-1}(w_t^{(m)} - w^{(t-1)}) + d_t^{(m)}$, $\delta_t := \frac{1}{|I_t|} \sum_{\xi \in I_t} \delta_t^{(\xi)}$ and $\bar{\delta}_t := \frac{1}{M} \sum_{m=1}^{M} \delta_t^{(m)}$. Then we have $\mathbb{E}[\delta_t] = \bar{\delta}_t$ and

$$w_t = w_{t-1} - \eta_t(d_t - \delta_t).$$

It can be verified based on Lemma 5 and triangle inequality that the following holds almost surely:

$$\max \left\{ \|\bar{\delta}_t\|, \|\delta_t\| \right\} \leq \sqrt{2(L + \eta_t^{-1})\varepsilon_t}. \qquad (11)$$

Since the loss is $L$-smooth, we can show that

$$\mathbb{E}[\bar{R}_{\mathrm{erm}}(w_t) \mid \mathcal{F}_{t-1}]$$

$$\leq \mathbb{E}\left[\bar{R}_{\mathrm{erm}}(w_{t-1}) + \left\langle \nabla \bar{R}_{\mathrm{erm}}(w_{t-1}), w_t - w_{t-1} \right\rangle + \frac{L}{2} \|w_t - w_{t-1}\|^2 \mid \mathcal{F}_{t-1}\right]$$

$$= \mathbb{E}\left[\bar{R}_{\mathrm{erm}}(w_{t-1}) - \eta_t \left\langle \nabla \bar{R}_{\mathrm{erm}}(w_{t-1}), d_t - \delta_t \right\rangle + \frac{L\eta_t^2}{2} \|d_t - \delta_t\|^2 \mid \mathcal{F}_{t-1}\right]$$

$$= \bar{R}_{\mathrm{erm}}(w_{t-1}) + \mathbb{E}\left[-\eta_t \left\langle \nabla \bar{R}_{\mathrm{erm}}(w_{t-1}), \bar{d}_t - \bar{\delta}_t \right\rangle + \frac{L\eta_t^2}{2} \|d_t - \delta_t\|^2 \mid \mathcal{F}_{t-1}\right]$$

$$\overset{\zeta_1}{\leq} \bar{R}_{\mathrm{erm}}(w_{t-1})$$
$$+ \mathbb{E}\left[-\eta_t \left\langle \nabla \bar{R}_{\mathrm{erm}}(w_{t-1}), \bar{d}_t \right\rangle + \eta_t G \|\bar{\delta}_t\| + \frac{3L\eta_t^2}{2} \|\bar{d}_t\|^2 + \frac{3L\eta_t^2}{2} \|d_t - \bar{d}_t\|^2 + \frac{3L\eta_t^2}{2} \|\delta_t\|^2 \mid \mathcal{F}_{t-1}\right]$$

$$\overset{\zeta_2}{\leq} \bar{R}_{\mathrm{erm}}(w_{t-1}) + \mathbb{E}\left[-\frac{\eta_t}{2} \|\nabla \bar{R}_{\mathrm{erm}}(w_{t-1})\|^2 - \frac{\eta_t}{2} \|\bar{d}_t\|^2 + \frac{\eta_t}{2} \|\nabla \bar{R}_{\mathrm{erm}}(w_{t-1}) - \bar{d}_t\|^2 + G\eta_t \sqrt{2(L + \eta_t^{-1})\varepsilon_t}\right.$$
$$\left. + \frac{3L\eta_t^2}{2} \|\bar{d}_t\|^2 + \frac{3LG^2\eta_t^2}{2I} + 3L(L + \eta_t^{-1})\eta_t^2 \varepsilon_t \mid \mathcal{F}_{t-1}\right]$$

$$\overset{\zeta_3}{\leq} \bar{R}_{\mathrm{erm}}(w_{t-1}) - \frac{\eta_t}{2} \|\nabla \bar{R}_{\mathrm{erm}}(w_{t-1})\|^2 + \mathbb{E}\left[\frac{\eta_t}{2} \|\nabla \bar{R}_{\mathrm{erm}}(w_{t-1}) - \bar{d}_t\|^2 \mid \mathcal{F}_{t-1}\right]$$
$$+ \frac{3LG^2\eta_t^2}{2I} + G\eta_t \sqrt{2(L + \eta_t^{-1})\varepsilon_t} + 3L(L + \eta_t^{-1})\eta_t^2 \varepsilon_t$$

$$\overset{\zeta_4}{\leq} \bar{R}_{\mathrm{erm}}(w_{t-1}) - \frac{\eta_t}{2} \|\nabla \bar{R}_{\mathrm{erm}}(w_{t-1})\|^2 + \frac{L^2 \left(G + \sqrt{2(L + \eta_t^{-1})\varepsilon_t}\right)^2 \eta_t^3}{2} + \frac{3LG^2\eta_t^2}{2I}$$
$$+ G\eta_t \sqrt{2(L + \eta_t^{-1})\varepsilon_t} + 3L(L + \eta_t^{-1})\eta_t^2 \varepsilon_t$$

$$\leq \bar{R}_{\mathrm{erm}}(w_{t-1}) - \frac{\eta_t}{2} \|\nabla \bar{R}_{\mathrm{erm}}(w_{t-1})\|^2 + 2L^2 G^2 \eta_t^3 + \frac{5LG^2\eta_t^2}{I},$$

where in "$\zeta_1$" we have used the $G$-Lipschitz of loss and triangle inequality, in "$\zeta_2$" we have used Lemma 4 and (11), in "$\zeta_3$" we have used $\eta_t \leq \frac{1}{3L}$, in "$\zeta_4$" we have used the first bound of Lemma 6, and in the last inequality we have used the condition of $\varepsilon_t \leq \min\left\{\frac{2L^2 G^2 \eta_t^3}{I^2(L\eta_t+1)}, \frac{G^2 \eta_t}{2I(L\eta_t+1)}\right\}$. Rearranging the terms and taking expectation over $\mathcal{F}_{t-1}$ in the above yields

$$\mathbb{E}\left[\|\nabla \bar{R}_{\mathrm{erm}}(w_{t-1})\|^2\right\} \leq \frac{2}{\eta_t} \mathbb{E}\left[\bar{R}_{\mathrm{erm}}(w_{t-1}) - \bar{R}_{\mathrm{erm}}(w_t)\right] + 4L^2 G^2 \eta_t^2 + \frac{10LG^2\eta_t}{I}.$$

Averaging the above from over $t = 1, 2, ..., T$ with $\eta_t \equiv \eta$ yields

$$\frac{1}{T} \sum_{t=0}^{T-1} \mathbb{E}\left[\|\nabla \bar{R}_{\mathrm{erm}}(w_t)\|^2\right] \leq \frac{2}{\eta T} \mathbb{E}\left[\bar{R}_{\mathrm{erm}}(w_0) - \bar{R}_{\mathrm{erm}}(w_T)\right] + 4L^2 G^2 \eta^2 + \frac{10LG^2\eta}{I}$$

$$\leq \frac{2}{\eta T} \bar{\Delta}_{\mathrm{erm}}^{(0)} + 4L^2 G^2 \eta^2 + \frac{10LG^2\eta}{I}.$$

If $T < I^3$, setting $\eta = \frac{1}{3LT^{1/3}}$ yields

$$\frac{1}{T} \sum_{t=0}^{T-1} \mathbb{E}\left[\|\nabla \bar{R}_{\mathrm{erm}}(w_t)\|^2\right] \lesssim \frac{L\bar{\Delta}_{\mathrm{erm}}^{(0)} + G^2}{T^{2/3}} + \frac{G^2}{T^{1/3}I} \lesssim \frac{L\bar{\Delta}_{\mathrm{erm}}^{(0)} + G^2}{T^{2/3}}.$$

If $T \geq I^3$, setting $\eta = \frac{1}{3L}\sqrt{\frac{I}{T}}$ yields

$$\frac{1}{T} \sum_{t=0}^{T-1} \mathbb{E}\left[\|\nabla \bar{R}_{\mathrm{erm}}(w_t)\|^2\right] \lesssim \frac{L\bar{\Delta}_{\mathrm{erm}}^{(0)} + G^2}{\sqrt{TI}} + \frac{G^2 I}{T} \lesssim \frac{L\bar{\Delta}_{\mathrm{erm}}^{(0)} + G^2}{\sqrt{TI}}.$$

Combining the preceding two inequalities and appealing to the definition of $w_{t^*}$ yields the desired bound. $\qquad\square$

## B.2 Proof of Theorem 2

We first present the following elementary lemma which will be used in the proof. It can be viewed as an inexact extension of the well-known three-point lemma to weakly convex functions.

**Lemma 7.** *Let $f$ be a $\nu$-weakly convex function and $\eta < \frac{1}{\nu}$. Consider*

$$w^+ = \arg\min_u \left\{ f(u) + \frac{1}{2\eta} \|u - w\|^2 \right\}.$$

*Then for any $u$, we have*

$$f(w^+) + \frac{1}{2\eta} \|w^+ - w\|^2 \le f(u) + \frac{1}{2\eta} \|u - w\|^2 - \frac{1/\eta - \nu}{2} \|w^+ - u\|^2.$$

*Proof.* Since $\eta < \frac{1}{\nu}$, we must have that the regularized objective $f(u) + \frac{1}{2\eta} \|u - w\|^2$ is $(1/\eta - \nu)$-strongly convex with respect to $u$, which immediately implies the desired bound. □

We will make use of the following lemma which shows that $w_t^{(m)}$ will be close to $w_{t-1}$ if the learning rate $\eta_t$ is small enough.

**Lemma 8.** *Assume that for each $m \in [M]$, the loss function $\ell^{(m)}$ is $G$-Lipschitz and $\nu$-weakly convex with respect to its first argument. Suppose that the local update oracle of `FedProx` is exactly solved and $\eta_t < \frac{1}{\nu}$. Then it holds that*

$$\left\| w_t^{(m)} - w_{t-1} \right\| \le G\eta_t.$$

*Proof.* Recall $Q_{\texttt{erm}}^{(m)}(w; w_{t-1}) = R_{\texttt{erm}}^{(m)}(w) + \frac{1}{2\eta_t} \|w - w_{t-1}\|^2$. Since the loss function is $\nu$-weakly convex and $\eta_t < \frac{1}{\nu}$, $Q_{\texttt{erm}}^{(m)}(w; w_{t-1})$ is strongly convex and thus admits a global minimizer. Since the local update oracle is exactly solved, we must have

$$\left\| \nabla R_{\texttt{erm}}^{(m)}(w_t^{(m)}) + \frac{1}{\eta_t}(w_t^{(m)} - w_{t-1}) \right\| = 0,$$

which implies the desired bound due to the $G$-Lipschitzness. □

With the above two preliminary lemmas in place, we are now in the position to prove the main result in Theorem 2.

*Proof of Theorem 2.* Since the losses are $\nu$-weakly convex and $\eta_t < \frac{1}{\nu}$, in view of Lemma 7 we can show for each $m \in [M]$ that the following holds for any $w$,

$$R_{\texttt{erm}}^{(m)}(w_t^{(m)}) + \frac{1}{2\eta_t} \|w_t^{(m)} - w_{t-1}\|^2 \le R_{\texttt{erm}}^{(m)}(w) + \frac{1}{2\eta_t} \|w - w_{t-1}\|^2 - \frac{1/\eta_t - \nu}{2} \|w_t^{(m)} - w\|^2. \quad (12)$$

Let us denote

$$\bar{w}_{t-1} := \text{prox}_{\rho \bar{R}_{\texttt{erm}}}(w_{t-1}) = \arg\min_w \left\{ \bar{R}_{\texttt{erm}}(w) + \frac{1}{2\rho} \|w - w_{t-1}\|^2 \right\}.$$

Setting $w = \bar{w}_{t-1}$ in the right hand side of (12) yields

$$R_{\texttt{erm}}^{(m)}(w_t^{(m)}) + \frac{1}{2\eta_t} \|w_t^{(m)} - w_{t-1}\|^2 \le R_{\texttt{erm}}^{(m)}(\bar{w}_{t-1}) + \frac{1}{2\eta_t} \|\bar{w}_{t-1} - w_{t-1}\|^2 - \frac{1/\eta_t - \nu}{2} \|w_t^{(m)} - \bar{w}_{t-1}\|^2.$$

In view of the above inequality we can show that for any $\xi \in I_t$,

$$R_{\mathrm{erm}}^{(\xi)}(w_{t-1}) + \frac{1}{2\eta_t}\|w_t^{(\xi)} - w_{t-1}\|^2$$

$$=R_{\mathrm{erm}}^{(\xi)}(w_t^{(\xi)}) + \frac{1}{2\eta_t}\|w_t^{(\xi)} - w_{t-1}\|^2 + R_{\mathrm{erm}}^{(\xi)}(w_{t-1}) - R_{\mathrm{erm}}^{(\xi)}(w_t^{(\xi)})$$

$$\leq R_{\mathrm{erm}}^{(\xi)}(w_t^{(\xi)}) + \frac{1}{2\eta_t}\|w_t^{(\xi)} - w_{t-1}\|^2 + G\|w_{t-1} - w_t^{(\xi)}\| \qquad (13)$$

$$\leq R_{\mathrm{erm}}^{(\xi)}(w_t^{(\xi)}) + \frac{1}{2\eta_t}\|w_t^{(\xi)} - w_{t-1}\|^2 + G^2\eta_t$$

$$\leq R_{\mathrm{erm}}^{(\xi)}(\bar{w}_{t-1}) + \frac{1}{2\eta_t}\|\bar{w}_{t-1} - w_{t-1}\|^2 - \frac{1/\eta_t - \nu}{2}\|w_t^{(\xi)} - \bar{w}_{t-1}\|^2 + G^2\eta_t,$$

where in the last but one inequality we have applied Lemma 8. Now recall that $w_t = \frac{1}{I}\sum_{\xi \in I_t} w_t^{(\xi)}$. Then based on triangle inequality we can see that

$$\frac{1}{I}\sum_{\xi \in I_t} R_{\mathrm{erm}}^{(\xi)}(w_{t-1}) + \frac{1}{2\eta_t}\|w_t - w_{t-1}\|^2$$

$$=\frac{1}{I}\sum_{\xi \in I_t} R_{\mathrm{erm}}^{(\xi)}(w_{t-1}) + \frac{1}{2\eta_t}\left\|\frac{1}{I}\sum_{\xi \in I_t} w_t^{(\xi)} - w_{t-1}\right\|^2$$

$$\leq \frac{1}{I}\sum_{\xi \in I_t}\left\{ R_{\mathrm{erm}}^{(\xi)}(w_{t-1}) + \frac{1}{2\eta_t}\left\|w_t^{(\xi)} - w_{t-1}\right\|^2 \right\}$$

$$\overset{(13)}{\leq} \frac{1}{I}\sum_{\xi \in I_t}\left\{ R_{\mathrm{erm}}^{(\xi)}(\bar{w}_{t-1}) + \frac{1}{2\eta_t}\|\bar{w}_{t-1} - w_{t-1}\|^2 - \frac{1/\eta_t - \nu}{2}\|w_t^{(\xi)} - \bar{w}_{t-1}\|^2 + G^2\eta_t \right\}$$

$$\leq \frac{1}{I}\sum_{\xi \in I_t} R_{\mathrm{erm}}^{(\xi)}(\bar{w}_{t-1}) + \frac{1}{2\eta_t}\|\bar{w}_{t-1} - w_{t-1}\|^2 - \frac{1/\eta_t - \nu}{2}\left\|\frac{1}{I}\sum_{\xi \in I_t} w_t^{(\xi)} - \bar{w}_{t-1}\right\|^2 + G^2\eta_t$$

$$=\frac{1}{I}\sum_{\xi \in I_t} R_{\mathrm{erm}}^{(\xi)}(\bar{w}_{t-1}) + \frac{1}{2\eta_t}\|\bar{w}_{t-1} - w_{t-1}\|^2 - \frac{1/\eta_t - \nu}{2}\|w_t - \bar{w}_{t-1}\|^2 + G^2\eta_t.$$

Conditioned on $\mathcal{F}_{t-1}$, taking expectation over both sides of the above inequality leads to the following inequality:

$$\mathbb{E}\left[ \bar{R}_{\mathrm{erm}}(w_{t-1}) + \frac{1}{2\eta_t}\|w_t - w_{t-1}\|^2 \mid \mathcal{F}_{t-1} \right]$$

$$=\mathbb{E}\left[ \frac{1}{I}\sum_{\xi \in I_t} R_{\mathrm{erm}}^{(\xi)}(w_{t-1}) + \frac{1}{2\eta_t}\|w_t - w_{t-1}\|^2 \mid \mathcal{F}_{t-1} \right]$$

$$\leq \mathbb{E}\left[ \frac{1}{I}\sum_{\xi \in I_t} R_{\mathrm{erm}}^{(\xi)}(\bar{w}_{t-1}) + \frac{1}{2\eta_t}\|\bar{w}_{t-1} - w_{t-1}\|^2 - \frac{1/\eta_t - \nu}{2}\|w_t - \bar{w}_{t-1}\|^2 + G^2\eta_t \mid \mathcal{F}_{t-1} \right]$$

$$=\mathbb{E}\left[ \bar{R}_{\mathrm{erm}}(\bar{w}_{t-1}) + \frac{1}{2\eta_t}\|\bar{w}_{t-1} - w_{t-1}\|^2 - \frac{1/\eta_t - \nu}{2}\|w_t - \bar{w}_{t-1}\|^2 + G^2\eta_t \mid \mathcal{F}_{t-1} \right].$$

Based the above inequality and by applying Lemma 8 again we can show that

$$
\mathbb{E}\left[\bar{R}_{\mathrm{erm}}(w_t) + \frac{1}{2\eta_t}\|w_t - w_{t-1}\|^2 \mid \mathcal{F}_{t-1}\right]
$$

$$
=\mathbb{E}\left[\bar{R}_{\mathrm{erm}}(w_{t-1}) + \frac{1}{2\eta_t}\|w_t - w_{t-1}\|^2 + \bar{R}_{\mathrm{erm}}(w_t) - \bar{R}_{\mathrm{erm}}(w_{t-1}) \mid \mathcal{F}_{t-1}\right]
$$

$$
\leq\mathbb{E}\left[\bar{R}_{\mathrm{erm}}(\bar{w}_{t-1}) + \frac{1}{2\eta_t}\|\bar{w}_{t-1} - w_{t-1}\|^2 - \frac{1/\eta_t - \nu}{2}\|w_t - \bar{w}_{t-1}\|^2 + G\|w_t - w_{t-1}\| \mid \mathcal{F}_{t-1}\right]
$$

$$
\leq\mathbb{E}\left[\bar{R}_{\mathrm{erm}}(\bar{w}_{t-1}) + \frac{1}{2\eta_t}\|\bar{w}_{t-1} - w_{t-1}\|^2 - \frac{1/\eta_t - \nu}{2}\|w_t - \bar{w}_{t-1}\|^2 + 2G^2\eta_t \mid \mathcal{F}_{t-1}\right],
$$

$$\tag{14}$$

where in the last inequality we have used $\|w_t - w_{t-1}\| \leq \frac{1}{I}\sum_{\xi\in I_t}\|w_t^{(\xi)} - w_{t-1}\| \leq G\eta_t$ due to triangle inequality and Lemma 8.

Since $\bar{R}_{\mathrm{erm}}$ is also $\nu$-weakly convex, invoking Lemma 7 to $\bar{w}_{t-1} = \mathrm{prox}_{\rho\bar{R}_{\mathrm{erm}}}(w_{t-1})$ yields

$$
\bar{R}_{\mathrm{erm}}(\bar{w}_{t-1}) + \frac{1}{2\rho}\|\bar{w}_{t-1} - w_{t-1}\|^2 \leq \bar{R}_{\mathrm{erm}}(w_t) + \frac{1}{2\rho}\|w_t - w_{t-1}\|^2 - \frac{1/\rho - \nu}{2}\|\bar{w}_{t-1} - w_t\|^2,
$$

which immediately leads to the following conditioned expectation bound:

$$
\mathbb{E}\left[\bar{R}_{\mathrm{erm}}(\bar{w}_{t-1}) + \frac{1}{2\rho}\|\bar{w}_{t-1} - w_{t-1}\|^2 \mid \mathcal{F}_{t-1}\right]
$$

$$
\leq\mathbb{E}\left[\bar{R}_{\mathrm{erm}}(w_t) + \frac{1}{2\rho}\|w_t - w_{t-1}\|^2 - \frac{1/\rho - \nu}{2}\|\bar{w}_{t-1} - w_t\|^2 \mid \mathcal{F}_{t-1}\right].
$$

$$\tag{15}$$

By summing up (14) and (15) we have

$$
\mathbb{E}\left[\frac{1/\eta_t - 1/\rho}{2}\|w_t - w_{t-1}\|^2 \mid \mathcal{F}_{t-1}\right]
$$

$$
\leq\mathbb{E}\left[\frac{1/\eta_t - 1/\rho}{2}\|\bar{w}_{t-1} - w_{t-1}\|^2 - \frac{1/\eta_t + 1/\rho - 2\nu}{2}\|\bar{w}_{t-1} - w_t\|^2 + 2G^2\eta_t \mid \mathcal{F}_{t-1}\right].
$$

Since by assumption $\eta_t \leq \rho$, rearranging the terms in the above yields

$$
\mathbb{E}\left[\|w_t - \bar{w}_{t-1}\|^2 \mid \mathcal{F}_{t-1}\right]
$$

$$
\leq\frac{1/\eta_t - 1/\rho}{1/\eta_t + 1/\rho - 2\nu}\|\bar{w}_{t-1} - w_{t-1}\|^2 + \frac{4G^2\eta_t}{1/\eta_t + 1/\rho - 2\nu}
$$

$$
\leq\|\bar{w}_{t-1} - w_{t-1}\|^2 - \frac{2(1/\rho - \nu)}{1/\eta_t + 1/\rho - 2\nu}\|\bar{w}_{t-1} - w_{t-1}\|^2 + \frac{4G^2\eta_t}{1/\eta_t + 1/\rho - 2\nu}.
$$

Then based on the above and the definition of Moreau envelope we can show that

$$
\mathbb{E}\left[\bar{R}_{\mathrm{erm},\rho}(w_t) \mid \mathcal{F}_{t-1}\right]
$$

$$
=\mathbb{E}\left[\bar{R}_{\mathrm{erm}}(\bar{w}_t) + \frac{1}{2\rho}\|\bar{w}_t - w_t\|^2 \mid \mathcal{F}_{t-1}\right]
$$

$$
\leq\mathbb{E}\left[\bar{R}_{\mathrm{erm}}(\bar{w}_{t-1}) + \frac{1}{2\rho}\|\bar{w}_{t-1} - w_t\|^2 \mid \mathcal{F}_{t-1}\right]
$$

$$
=\bar{R}_{\mathrm{erm}}(\bar{w}_{t-1}) + \frac{1}{2\rho}\mathbb{E}\left[\|\bar{w}_{t-1} - w_t\|^2 \mid \mathcal{F}_{t-1}\right]
$$

$$
\leq\bar{R}_{\mathrm{erm}}(\bar{w}_{t-1}) + \frac{1}{2\rho}\|\bar{w}_{t-1} - w_{t-1}\|^2 - \frac{(1/\rho - \nu)/\rho}{1/\eta_t + 1/\rho - 2\nu}\|\bar{w}_{t-1} - w_{t-1}\|^2 + \frac{2G^2\eta_t/\rho}{1/\eta_t + 1/\rho - 2\nu}
$$

$$
=\bar{R}_{\mathrm{erm},\rho}(w_{t-1}) - \frac{(1/\rho - \nu)/\rho}{1/\eta_t + 1/\rho - 2\nu}\|\bar{w}_{t-1} - w_{t-1}\|^2 + \frac{2G^2\eta_t/\rho}{1/\eta_t + 1/\rho - 2\nu}
$$

$$
=\bar{R}_{\mathrm{erm},\rho}(w_{t-1}) - \frac{1 - \rho\nu}{1/\eta_t + 1/\rho - 2\nu}\left\|\nabla\bar{R}_{\mathrm{erm},\rho}(w_{t-1})\right\|^2 + \frac{2G^2\eta_t/\rho}{1/\eta_t + 1/\rho - 2\nu},
$$

where in the last equality we have used the identity $\|\bar{w}_{t-1} - w_{t-1}\|^2 = \rho^2 \|\nabla \bar{R}_{\mathrm{erm},\rho}(w_{t-1})\|^2$ (see, e.g., Davis and Drusvyatskiy, 2019). By rearranging the terms in the above and taking expectation over $\mathcal{F}_{t-1}$ we obtain that

$$\frac{1-\rho\nu}{1/\eta_t + 1/\rho - 2\nu} \mathbb{E}\left[\left\|\nabla \bar{R}_{\mathrm{erm},\rho}(w_{t-1})\right\|^2\right] \leq \mathbb{E}\left[\bar{R}_{\mathrm{erm},\rho}(w_{t-1})\right] - \mathbb{E}\left[\bar{R}_{\mathrm{erm},\rho}(w_t)\right] + \frac{2G^2\eta_t/\rho}{1/\eta_t + 1/\rho - 2\nu}.$$

Averaging the above over $t = 1, ..., T$ yields

$$\begin{aligned}
\frac{1}{T}\sum_{t=0}^{T-1}\mathbb{E}\left[\left\|\nabla \bar{R}_{\mathrm{erm},\rho}(w_t)\right\|^2\right] \leq & \frac{1/\eta_t + 1/\rho - 2\nu}{T(1-\rho\nu)}\mathbb{E}\left[\bar{R}_{\mathrm{erm},\rho}(w_0) - \bar{R}_{\mathrm{erm},\rho}(w_T)\right] + \frac{2G^2\eta_t}{\rho(1-\rho\nu)} \\
\leq & \frac{1/\eta_t + 1/\rho - 2\nu}{T(1-\rho\nu)}\bar{\Delta}_{\mathrm{erm},\rho}^{(0)} + \frac{2G^2\eta_t}{\rho(1-\rho\nu)} \\
= & \frac{(1-2\rho\nu)\bar{\Delta}_{\mathrm{erm},\rho}^{(0)}}{T\rho(1-\rho\nu)} + \frac{\bar{\Delta}_{\mathrm{erm},\rho}^{(0)}}{\eta_t T(1-\rho\nu)} + \frac{2G^2\eta_t}{\rho(1-\rho\nu)} \\
\leq & \frac{\bar{\Delta}_{\mathrm{erm},\rho}^{(0)}}{T\rho} + \frac{2\bar{\Delta}_{\mathrm{erm},\rho}^{(0)}}{\eta_t T} + \frac{4G^2\eta_t}{\rho} \\
= & \frac{\bar{\Delta}_{\mathrm{erm},\rho}^{(0)}}{T\rho} + \frac{2\bar{\Delta}_{\mathrm{erm},\rho}^{(0)} + 4G^2\rho}{\rho\sqrt{T}},
\end{aligned}$$

where in the last but one inequality we have used $\rho < \frac{1}{2\nu}$, and in the last inequality we have used the choice of $\eta_t \equiv \frac{\rho}{\sqrt{T}}$. The desired bound follows by preserving the dominant terms in the above bound and appealing to the definition of $t^*$. $\qquad\square$

## C   Proofs for Section 3

### C.1   Proof of Theorem 3

For each time instance $t$, let us overload the notation $d_t^{(m)}$ as

$$d_t^{(m)} = \nabla R_{B_t^{(m)}}^{(m)}(w_t^{(m)}) = \frac{1}{b}\sum_{i=1}^{b}\nabla \ell^{(m)}(w_t^{(m)}; z_{i,t}^{(m)}).$$

We then accordingly overload the quantities $d_t$ and $\bar{d}_t$ as defined in (9). We then have the following lemma analogous to Lemma 5.

**Lemma 9.** *Assume that for each $m \in [M]$, the loss function $\ell^{(m)}$ is $L$-smooth with respect to its first argument. Suppose that the local update oracle of* FedMSPP *is $\varepsilon_t$-inexactly solved and $\eta_t < \frac{1}{L}$. Then it holds that*

$$\left\|w_t^{(m)} - w_{t-1} + \eta_t d_t^{(m)}\right\| \leq \eta_t\sqrt{2(L + \eta_t^{-1})\varepsilon_t}.$$

*Proof.* Consider $Q_{B_t^{(m)}}^{(m)}(w; w_{t-1}) = R_{B_t^{(m)}}^{(m)}(w) + \frac{1}{2\eta_t}\|w - w_{t-1}\|^2$. Since the loss functions are $L$-smooth and $\eta_t < \frac{1}{L}$, $Q_{B_t^{(m)}}^{(m)}(w; w_{t-1})$ must be strongly convex and thus admits a global minimizer. Then we have

$$\begin{aligned}
& \left\|\nabla R_{B_t^{(m)}}^{(m)}(w_t^{(m)}) + \frac{1}{\eta_t}(w_t^{(m)} - w_{t-1})\right\|^2 \\
= & \left\|\nabla Q_{B_t^{(m)}}^{(m)}(w_t^{(m)}; w_{t-1})\right\|^2 \\
\leq & 2(L + \eta_t^{-1})\left(Q_{B_t^{(m)}}^{(m)}(w_t^{(m)}; w_{t-1}) - \min_w Q_{B_t^{(m)}}^{(m)}(w; w_{t-1})\right) \leq 2(L + \eta_t^{-1})\varepsilon_t,
\end{aligned}$$

where in the last inequality is due to Definition 3. This implies the desired bound. $\qquad\square$

Let $\{\mathcal{F}_t\}_{t\geq 1}$ be the filtration generated by the random iterates $\{w_t\}_{t\geq 1}$ as $\mathcal{F}_t = \sigma\left(w_1, w_2, ..., w_t\right)$, where the randomness jointly comes from the sampling of devices for partial participation and sampling of minibatch for local update on each chosen device.

**Lemma 10.** *Assume that for each $m \in [M]$, the loss function $\ell^{(m)}$ is G-Lipschitz and L-smooth with respect to its first argument. Suppose that $\eta_t < \frac{1}{L}$ and the local update oracle of* `FedMSPP` *is $\varepsilon_t$-inexactly solved with $\varepsilon_t \leq \frac{G^2 \eta_t}{8b^2}$. Then it holds for every $m \in [M]$ that*

$$\left\| \mathbb{E}\left[ \nabla R^{(m)}(w_t^{(m)}) - d_t^{(m)} \mid \mathcal{F}_{t-1} \right] \right\| \leq \frac{5LG\eta_t}{(1 - \eta_t L)b},$$

$$\mathbb{E}\left[ \left\| \nabla R^{(m)}(w_t^{(m)}) - \mathbb{E}[\nabla R^{(m)}(w_t^{(m)}) \mid \mathcal{F}_{t-1}] \right\|^2 \mid \mathcal{F}_{t-1} \right] \leq \frac{25L^2 G^2 \eta_t}{(1 - \eta_t L)^2 b}.$$

*Proof.* Let us recall Definition 3 where the inexact solution $w_t^{(m)}$ is given by

$$Q_{B_t^{(m)}}^{(m)}(w_t^{(m)}; w_{t-1}) \leq \min_w Q_{B_t^{(m)}}^{(m)}(w; w_{t-1}) + \varepsilon_t.$$

Since the loss functions are $L$-smooth and $\frac{1}{\eta_t} > L$, it is easy to verify that the regularized objective $Q_{B_t^{(m)}}^{(m)}(w; w_{t-1})$ is $(\frac{1}{\eta_t} - L)$-strongly convex. Then invoking Lemma 1 yields that $w_t^{(m)}$ uniformly stable with parameter $\frac{4G}{(1/\eta_t - L)b} + 2\sqrt{\frac{2\varepsilon_t}{1/\eta_t - L}} \leq \frac{5G}{(1/\eta_t - L)b}$, which is due to the condition on $\varepsilon_t$. Conditioned on the sigma-field $\mathcal{F}_{t-1}$, the desired bounds follows immediately from Lemma 3. $\square$

The next lemma, which can be proved based on the previous lemmas, is key to our analysis.

**Lemma 11.** *Assume that for each $m \in [M]$, the loss function $\ell^{(m)}$ is G-Lipschitz and L-smooth with respect to its first argument. Suppose that $\eta_t < \frac{1}{L}$ and the local update oracle of* `FedMSPP` *is $\varepsilon_t$-inexactly solved with $\varepsilon_t \leq \min\left\{ \frac{G^2 \eta_t}{2(L\eta_t + 1)}, \frac{G^2 \eta_t}{8b^2} \right\}$. Then we have*

$$\mathbb{E}\left[ \left\| \nabla \bar{R}(w_{t-1}) - d_t \right\|^2 \mid \mathcal{F}_{t-1} \right] \leq 8L^2 G^2 \eta_t^2 + \frac{2G^2}{b|I_t|},$$

*and*

$$\left\| \nabla \bar{R}(w_{t-1}) - \mathbb{E}[\bar{d}_t \mid \mathcal{F}_{t-1}] \right\|^2 \leq 12L^2 G^2 \eta_t^2 + \frac{75 L^2 G^2 \eta_t^2}{(1 - \eta_t L)^2 b^2} + \frac{75 L^2 G^2 \eta_t^2}{(1 - \eta_t L)^2 b}.$$

*Proof.* By Lemma 9 we know that for each $m \in [M]$,

$$\|w_t^{(m)} - w_{t-1}\| \leq \eta_t \|d_t^{(m)}\| + \eta_t \sqrt{2(L + \eta_t^{-1})\varepsilon_t} \leq \left( G + \sqrt{2(L + \eta_t^{-1})\varepsilon_t} \right) \eta_t \leq 2G\eta_t, \quad (16)$$

where we have used the $G$-Lipschitz assumption of loss and $\varepsilon_t \leq \frac{G^2 \eta_t}{2(L\eta_t + 1)}$. By definition we can see that

$$\mathbb{E}\left[\|\nabla \bar{R}(w_{t-1}) - d_t\|^2 \mid \mathcal{F}_{t-1}\right]$$

$$= \mathbb{E}\left[\left\|\nabla \bar{R}(w_{t-1}) - \frac{1}{|I_t|}\sum_{\xi \in I_t} \nabla R_{B_t^{(\xi)}}^{(\xi)}(w_t^{(\xi)})\right\|^2 \mid \mathcal{F}_{t-1}\right]$$

$$= \mathbb{E}\left[\left\|\nabla \bar{R}(w_{t-1}) - \frac{1}{|I_t|}\sum_{\xi \in I_t} \nabla R_{B_t^{(\xi)}}^{(\xi)}(w_{t-1}) + \frac{1}{|I_t|}\sum_{\xi \in I_t} \nabla R_{B_t^{(\xi)}}^{(\xi)}(w_{t-1}) - \frac{1}{|I_t|}\sum_{\xi \in I_t} \nabla R_{B_t^{(\xi)}}^{(\xi)}(w_t^{(\xi)})\right\|^2 \mid \mathcal{F}_{t-1}\right]$$

$$\leq \mathbb{E}\left[2\left\|\nabla \bar{R}(w_{t-1}) - \frac{1}{b|I_t|}\sum_{\xi \in I_t}\sum_{i \in [b]} \nabla \ell^{(\xi)}(w_{t-1}; z_{i,t}^{(\xi)})\right\|^2 + \frac{2}{|I_t|}\sum_{\xi \in I_t}\left\|\nabla R_{B_t^{(\xi)}}^{(\xi)}(w_{t-1}) - \nabla R_{B_t^{(\xi)}}^{(\xi)}(w_t^{(\xi)})\right\|^2 \mid \mathcal{F}_{t-1}\right]$$

$$\overset{\zeta_1}{\leq} \frac{2}{b^2|I_t|^2}\sum_{\xi \in I_t}\sum_{i \in [b]}\mathbb{E}\left[\left\|\nabla \bar{R}(w_{t-1}) - \nabla \ell^{(\xi)}(w_{t-1}; z_{i,t}^{(\xi)})\right\|^2 \mid \mathcal{F}_{t-1}\right] + \frac{2L^2}{|I_t|}\sum_{\xi \in I_t}\mathbb{E}\left[\left\|w_{t-1} - w_t^{(\xi)}\right\|^2 \mid \mathcal{F}_{t-1}\right]$$

$$\overset{\zeta_2}{\leq} \frac{2}{b^2|I_t|^2}\sum_{\xi \in I_t}\sum_{i \in [b]}\mathbb{E}\left[\left\|\nabla \ell^{(\xi)}(w_{t-1}; z_{i,t}^{(\xi)})\right\|^2 \mid \mathcal{F}_{t-1}\right] + 8L^2 G^2 \eta_t^2$$

$$\leq \frac{2G^2}{b|I_t|} + 8L^2 G^2 \eta_t^2,$$

where in "$\zeta_1$" we have used the independent sampling of data and devices and the $L$-smoothness of loss, in "$\zeta_2$" we have used the fact $\mathbb{E}[\|Z - \mathbb{E}[Z]\|^2] \leq \mathbb{E}[\|Z\|^2]$ and (16), and in the last inequality we have used the $G$-Lipschitzness of loss . This proves the first desired bound.

To prove the second bound, by definition we can see that

$$\left\|\nabla \bar{R}(w_{t-1}) - \mathbb{E}[\bar{d}_t \mid \mathcal{F}_{t-1}]\right\|^2$$

$$= \left\|\frac{1}{M}\sum_{m=1}^{M}\left(\nabla R^{(m)}(w_{t-1}) - \mathbb{E}[d_t^{(m)} \mid \mathcal{F}_{t-1}]\right)\right\|^2$$

$$= \left\|\frac{1}{M}\sum_{m=1}^{M}\left(\nabla R^{(m)}(w_{t-1}) - \nabla R^{(m)}(w_t^{(m)}) + \nabla R^{(m)}(w_t^{(m)}) - \mathbb{E}[\nabla R^{(m)}(w_t^{(m)}) \mid \mathcal{F}_{t-1}]\right.\right.$$

$$\left.\left. + \mathbb{E}\left[\nabla R^{(m)}(w_t^{(m)}) - d_t^{(m)} \mid \mathcal{F}_{t-1}\right]\right)\right\|^2 .$$

$$\leq \underbrace{\frac{3}{M}\sum_{m=1}^{M}\left\|\nabla R^{(m)}(w_{t-1}) - \nabla R^{(m)}(w_t^{(m)})\right\|^2}_{A'} + \underbrace{\frac{3}{M}\sum_{m=1}^{M}\left\|\mathbb{E}\left[\nabla R^{(m)}(w_t^{(m)}) - d_t^{(m)} \mid \mathcal{F}_{t-1}\right]\right\|^2}_{B'}$$

$$+ \underbrace{\frac{3}{M}\sum_{m=1}^{M}\left\|\nabla R^{(m)}(w_t^{(m)}) - \mathbb{E}[\nabla R^{(m)}(w_t^{(m)}) \mid \mathcal{F}_{t-1}]\right\|^2}_{C'} .$$

By smoothness and (16) we can show that the following holds almost surely:

$$A' \leq 3L^2\|w_t^{(m)} - w_{t-1}\|^2 \leq 12L^2 G^2 \eta_t^2.$$

For the component $B'$, based on the first bound of Lemma 10 we can easily show that

$$B' \leq \frac{75L^2 G^2}{(1/\eta_t - L)^2 b^2}.$$

In terms of the component $C'$, it can be bounded via invoking the second bound of Lemma 10 that

$$\mathbb{E}\left[C' \mid \mathcal{F}_{t-1}\right] \leq \frac{75L^2 G^2}{(1/\eta_t - L)^2 b}.$$

Finally, by combing the preceding three bounds we obtain that

$$\left\|\nabla\bar{R}(w_{t-1}) - \mathbb{E}[\bar{d}_t \mid \mathcal{F}_{t-1}]\right\|^2 = \mathbb{E}\left[\left\|\nabla\bar{R}(w_{t-1}) - \mathbb{E}[\bar{d}_t \mid \mathcal{F}_{t-1}]\right\|^2 \mid \mathcal{F}_{t-1}\right]$$

$$\leq 12L^2G^2\eta_t^2 + \frac{75L^2G^2}{(1/\eta_t - L)^2b^2} + \frac{75L^2G^2}{(1/\eta_t - L)^2b}.$$

This proves the second desired bound. $\qquad\square$

With all the above preliminary results in place, we are now ready to prove Theorem 3.

*Proof of Theorem 3.* Let us denote $\delta_t^{(m)} := \eta_t^{-1}(w_t^{(m)} - w^{(t-1)}) + d_t^{(m)}$, $\delta_t := \frac{1}{|I_t|}\sum_{\xi\in I_t}\delta_t^{(\xi)}$ and $\bar{\delta}_t := \frac{1}{M}\sum_{m=1}^{M}\delta_t^{(m)}$. Then we have $\mathbb{E}[\delta_t] = \bar{\delta}_t$ and $w_t = w_{t-1} - \eta_t(d_t - \delta_t)$. It can be verified based on Lemma 9 and triangle inequality that the following holds almost surely:

$$\max\left\{\|\bar{\delta}_t\|, \|\delta_t\|\right\} \leq \sqrt{2(L + \eta_t^{-1})\varepsilon_t}. \tag{17}$$

Since the objective is $L$-smooth, we can show that

$$\mathbb{E}\left[\bar{R}(w_t) \mid \mathcal{F}_{t-1}\right]$$

$$\leq \mathbb{E}\left[\bar{R}(w_{t-1}) + \left\langle\nabla\bar{R}(w_{t-1}), w_t - w_{t-1}\right\rangle + \frac{L}{2}\|w_t - w_{t-1}\|^2 \mid \mathcal{F}_{t-1}\right]$$

$$= \mathbb{E}\left[\bar{R}(w_{t-1}) - \eta_t\left\langle\nabla\bar{R}(w_{t-1}), d_t - \delta_t\right\rangle + \frac{L\eta_t^2}{2}\|d_t - \delta_t\|^2 \mid \mathcal{F}_{t-1}\right]$$

$$= \mathbb{E}\left[\bar{R}(w_{t-1}) - \eta_t\left\langle\nabla\bar{R}(w_{t-1}), \mathbb{E}[\bar{d}_t - \bar{\delta}_t \mid \mathcal{F}_{t-1}]\right\rangle + \frac{L\eta_t^2}{2}\|d_t - \delta_t\|^2 \mid \mathcal{F}_{t-1}\right]$$

$$\leq \bar{R}(w_{t-1}) - \eta_t\left\langle\nabla\bar{R}(w_{t-1}), \mathbb{E}[\bar{d}_t \mid \mathcal{F}_{t-1}]\right\rangle + \mathbb{E}\left[\eta_tG\|\bar{\delta}_t\| + L\eta_t^2\|d_t\|^2 + L\eta_t^2\|\delta_t\|^2 \mid \mathcal{F}_{t-1}\right]$$

$$= \bar{R}(w_{t-1}) - \frac{\eta_t}{2}\left\|\nabla\bar{R}(w_{t-1})\right\|^2 - \frac{\eta_t}{2}\left\|\mathbb{E}[\bar{d}_t \mid \mathcal{F}_{t-1}]\right\|^2 + \frac{\eta_t}{2}\left\|\nabla\bar{R}(w_{t-1}) - \mathbb{E}[\bar{d}_t \mid \mathcal{F}_{t-1}]\right\|^2$$

$$\quad + \mathbb{E}\left[\eta_tG\|\bar{\delta}_t\| + L\eta_t^2\|d_t\|^2 + L\eta_t^2\|\delta_t\|^2 \mid \mathcal{F}_{t-1}\right]$$

$$\overset{\zeta_1}{\leq} \bar{R}(w_{t-1}) - \frac{\eta_t}{2}\left\|\nabla\bar{R}(w_{t-1})\right\|^2 + \frac{\eta_t}{2}\left\|\nabla\bar{R}(w_{t-1}) - \mathbb{E}[\bar{d}_t \mid \mathcal{F}_{t-1}]\right\|^2$$

$$\quad + \mathbb{E}\left[G\eta_t\sqrt{2(L + \eta_t^{-1})\varepsilon_t} + L\eta_t^2\|d_t\|^2 + 2L(L + \eta_t^{-1})\eta_t^2\varepsilon_t \mid \mathcal{F}_{t-1}\right]$$

$$\leq \bar{R}(w_{t-1}) - \frac{\eta_t}{2}\|\nabla\bar{R}(w_{t-1})\|^2 + \frac{\eta_t}{2}\|\nabla\bar{R}(w_{t-1}) - \mathbb{E}[\bar{d}_t \mid \mathcal{F}_{t-1}]\|^2$$

$$\quad + \mathbb{E}\left[2L\eta_t^2\left\|d_t - \nabla\bar{R}(w_{t-1})\right\|^2 + 2L\eta_t^2\left\|\nabla\bar{R}(w_{t-1})\right\|^2 + G\eta_t\sqrt{2(L + \eta_t^{-1})\varepsilon_t} + 2L(L + \eta_t^{-1})\eta_t^2\varepsilon_t \mid \mathcal{F}_{t-1}\right]$$

$$\overset{\zeta_2}{\leq} \bar{R}(w_{t-1}) - \frac{\eta_t}{4}\left\|\nabla\bar{R}(w_{t-1})\right\|^2 + \frac{\eta_t}{2}\left\|\nabla\bar{R}(w_{t-1}) - \mathbb{E}[\bar{d}_t \mid \mathcal{F}_{t-1}]\right\|^2$$

$$\quad + \mathbb{E}\left[2L\eta_t^2\left\|d_t - \nabla\bar{R}(w_{t-1})\right\|^2 \mid \mathcal{F}_{t-1}\right] + G\eta_t\sqrt{2(L + \eta_t^{-1})\varepsilon_t} + 2L(L + \eta_t^{-1})\eta_t^2\varepsilon_t$$

$$\overset{\zeta_3}{\leq} \bar{R}(w_{t-1}) - \frac{\eta_t}{4}\left\|\nabla\bar{R}(w_{t-1})\right\|^2 + 6L^2G^2\eta_t^3 + \frac{38L^2G^2\eta_t^3}{(1 - \eta_tL)^2b^2} + \frac{38L^2G^2\eta_t^3}{(1 - \eta_tL)^2b}$$

$$\quad + 16L^3G^2\eta_t^4 + \frac{4LG^2\eta_t^2}{bI} + G\eta_t\sqrt{2(L + \eta_t^{-1})\varepsilon_t} + 2L(L + \eta_t^{-1})\eta_t^2\varepsilon_t$$

$$\overset{\zeta_4}{\leq} \bar{R}(w_{t-1}) - \frac{\eta_t}{4}\left\|\nabla\bar{R}(w_{t-1})\right\|^2 + 6L^2G^2\eta_t^3 + \frac{43L^2G^2\eta_t^3}{b^2} + \frac{43L^2G^2\eta_t^3}{b}$$

$$\quad + 16L^3G^2\eta_t^4 + \frac{4LG^2\eta_t^2}{bI} + G\eta_t\sqrt{2(L + \eta_t^{-1})\varepsilon_t} + 2L(L + \eta_t^{-1})\eta_t^2\varepsilon_t$$

$$\overset{\zeta_5}{\leq} \bar{R}(w_{t-1}) - \frac{\eta_t}{4}\left\|\nabla\bar{R}(w_{t-1})\right\|^2 + 94L^2G^2\eta_t^3 + \frac{4LG^2\eta_t^2}{bI} + G\eta_t\sqrt{2(L + \eta_t^{-1})\varepsilon_t} + 2L(L + \eta_t^{-1})\eta_t^2\varepsilon_t$$

$$\leq \bar{R}(w_{t-1}) - \frac{\eta_t}{4}\left\|\nabla\bar{R}(w_{t-1})\right\|^2 + 94L^2G^2\eta_t^3 + \frac{6LG^2\eta_t^2}{bI},$$

where in "$\zeta_1$" we have used (17), in "$\zeta_2$" we have used $\eta_t \leq \frac{1}{8L}$, in "$\zeta_3$" we have used Lemma 11, in "$\zeta_4$" we have used $\eta_t \leq \frac{1}{8L}$, in "$\zeta_5$" we have used $M, b \geq 1$ and $\eta_t \leq \frac{1}{8L}$, and in the last inequality we used $\varepsilon_t \leq \frac{L^2 G^2 \eta_t^3}{2bI(L\eta_t+1)}$ and $\eta_t \leq \frac{1}{8L}$. By taking expectation over $\mathcal{F}_{t-1}$ and rearranging the terms we obtain that

$$\mathbb{E}\left[\left\|\nabla \bar{R}(w_{t-1})\right\|^2\right] \leq \frac{4}{\eta_t}\left(\mathbb{E}[\bar{R}(w_{t-1})] - \mathbb{E}[\bar{R}(w_t)]\right) + 376 L^2 G^2 \eta_t^2 + \frac{24 L G^2 \eta_t}{bI}.$$

Averaging the above from over $t = 1, 2, .., T$ with $\eta_t \equiv \eta$ yields

$$\frac{1}{T}\sum_{t=0}^{T-1}\mathbb{E}\left[\left\|\nabla \bar{R}(w_t)\right\|^2\right] \leq \frac{4}{\eta T}(\bar{R}(w_0) - \bar{R}(w_T)) + 376 L^2 G^2 \eta^2 + \frac{24 L \eta G^2}{bI}$$

$$\leq \frac{4\bar{\Delta}^{(0)}}{\eta T} + 376 L^2 G^2 \eta^2 + \frac{24 L \eta G^2}{bI}.$$

If $T < (bI)^3$, setting $\eta = \frac{1}{8LT^{1/3}}$ yields

$$\frac{1}{T}\sum_{t=0}^{T-1}\mathbb{E}\left[\|\nabla \bar{R}(w_t)\|^2\right] \lesssim \frac{L\bar{\Delta}^{(0)} + G^2}{T^{2/3}} + \frac{G^2}{T^{1/3}bI} \lesssim \frac{L\bar{\Delta}^{(0)} + G^2}{T^{2/3}}.$$

If $T \geq (bI)^3$, setting $\eta = \frac{1}{8L}\sqrt{\frac{bI}{T}}$ yields

$$\frac{1}{T}\sum_{t=0}^{T-1}\mathbb{E}\left[\|\nabla \bar{R}(w_t)\|^2\right] \lesssim \frac{L\bar{\Delta}^{(0)} + G^2}{\sqrt{TbI}} + \frac{G^2 bI}{T} \lesssim \frac{L\bar{\Delta}^{(0)} + G^2}{\sqrt{TbI}}.$$

Combining the preceding two inequalities and appealing to the definition of $t^*$ yield the desired bound. $\qquad\square$

## C.2   Proof of Theorem 4

The proof argument is almost identical to that of Theorem 2. We reproduce the proof in full details here for the sake of completeness.

Similar to Lemma 8, we first establish the following lemma which shows that $w_t^{(m)}$ will be close to $w_{t-1}$ if the learning rate $\eta_t$ is small enough.

**Lemma 12.** *Assume that for each $m \in [M]$, the loss function $\ell^{(m)}$ is $G$-Lipschitz and $\nu$-weakly convex with respect to its first argument. Suppose that the local update oracle of* `FedMSPP` *is exactly solved and $\eta_t < \frac{1}{\nu}$. Then it holds that*

$$\left\|w_t^{(m)} - w_{t-1}\right\| \leq G\eta_t.$$

*Proof.* Recall $Q_{B_t^{(m)}}^{(m)}(w; w_{t-1}) = R_{B_t^{(m)}}^{(m)}(w) + \frac{1}{2\eta_t}\|w - w_{t-1}\|^2$. Since the loss function is $\nu$-weakly convex and $\eta_t < \frac{1}{\nu}$, $Q_{B_t^{(m)}}^{(m)}(w; w_{t-1})$ is strongly convex with respect to $w$ and thus admits a global minimizer. Since the local update oracle is exactly solved, we must have

$$\left\|\nabla R_{B_t^{(m)}}^{(m)}(w_t^{(m)}) + \frac{1}{\eta_t}(w_t^{(m)} - w_{t-1})\right\| = 0,$$

which implies the desired bound due to the $G$-Lipschitz-loss assumption. $\qquad\square$

We are now ready to prove the main result in Theorem 4.

*Proof of Theorem 4.* Since the losses are $\nu$-weakly convex and $\eta_t < \frac{1}{\nu}$, in view of Lemma 7 we can show for each $m \in [M]$ that the following holds for any $w$,

$$R_{B_t^{(m)}}^{(m)}(w_t^{(m)}) + \frac{1}{2\eta_t}\|w_t^{(m)} - w_{t-1}\|^2 \leq R_{B_t^{(m)}}^{(m)}(w) + \frac{1}{2\eta_t}\|w - w_{t-1}\|^2 - \frac{1/\eta_t - \nu}{2}\|w_t^{(m)} - w\|^2.$$
$$\tag{18}$$

Let us denote for any $t \geq 1$

$$\bar{w}_{t-1} := \text{prox}_{\rho \bar{R}}(w_{t-1}) = \arg\min_w \left\{ \bar{R}(w) + \frac{1}{2\rho} \|w - w_{t-1}\|^2 \right\}.$$

Setting $w = \bar{w}_{t-1}$ in the right hand side of (18) yields

$$R_{B_t^{(m)}}^{(m)}(w_t^{(m)}) + \frac{1}{2\eta_t} \|w_t^{(m)} - w_{t-1}\|^2 \leq R_{B_t^{(m)}}^{(m)}(\bar{w}_{t-1}) + \frac{1}{2\eta_t} \|\bar{w}_{t-1} - w_{t-1}\|^2 - \frac{1/\eta_t - \nu}{2} \|w_t^{(m)} - \bar{w}_{t-1}\|^2.$$

In view of the above inequality we can show that for any $\xi \in I_t$,

$$R_{B_t^{(\xi)}}^{(\xi)}(w_{t-1}) + \frac{1}{2\eta_t} \|w_t^{(\xi)} - w_{t-1}\|^2$$

$$= R_{B_t^{(\xi)}}^{(\xi)}(w_t^{(\xi)}) + \frac{1}{2\eta_t} \|w_t^{(\xi)} - w_{t-1}\|^2 + R_{B_t^{(\xi)}}^{(\xi)}(w_{t-1}) - R_{B_t^{(\xi)}}^{(\xi)}(w_t^{(\xi)})$$

$$\leq R_{B_t^{(\xi)}}^{(\xi)}(w_t^{(\xi)}) + \frac{1}{2\eta_t} \|w_t^{(\xi)} - w_{t-1}\|^2 + G\|w_{t-1} - w_t^{(\xi)}\| \qquad (19)$$

$$\leq R_{B_t^{(\xi)}}^{(\xi)}(w_t^{(\xi)}) + \frac{1}{2\eta_t} \|w_t^{(\xi)} - w_{t-1}\|^2 + G^2 \eta_t$$

$$\leq R_{B_t^{(\xi)}}^{(\xi)}(\bar{w}_{t-1}) + \frac{1}{2\eta_t} \|\bar{w}_{t-1} - w_{t-1}\|^2 - \frac{1/\eta_t - \nu}{2} \|w_t^{(\xi)} - \bar{w}_{t-1}\|^2 + G^2 \eta_t,$$

where in the last but one inequality we have applied Lemma 12. Now recall that $w_t = \frac{1}{I} \sum_{\xi \in I_t} w_t^{(\xi)}$. Then based on triangle inequality we can see that

$$\frac{1}{I} \sum_{\xi \in I_t} R_{B_t^{(\xi)}}^{(\xi)}(w_{t-1}) + \frac{1}{2\eta_t} \|w_t - w_{t-1}\|^2$$

$$= \frac{1}{I} \sum_{\xi \in I_t} R_{B_t^{(\xi)}}^{(\xi)}(w_{t-1}) + \frac{1}{2\eta_t} \left\| \frac{1}{I} \sum_{\xi \in I_t} w_t^{(\xi)} - w_{t-1} \right\|^2$$

$$\leq \frac{1}{I} \sum_{\xi \in I_t} \left\{ R_{B_t^{(\xi)}}^{(\xi)}(w_{t-1}) + \frac{1}{2\eta_t} \left\| w_t^{(\xi)} - w_{t-1} \right\|^2 \right\}$$

$$\overset{(19)}{\leq} \frac{1}{I} \sum_{\xi \in I_t} \left\{ R_{B_t^{(\xi)}}^{(\xi)}(\bar{w}_{t-1}) + \frac{1}{2\eta_t} \|\bar{w}_{t-1} - w_{t-1}\|^2 - \frac{1/\eta_t - \nu}{2} \|w_t^{(\xi)} - \bar{w}_{t-1}\|^2 + G^2 \eta_t \right\}$$

$$\leq \frac{1}{I} \sum_{\xi \in I_t} R_{B_t^{(\xi)}}^{(\xi)}(\bar{w}_{t-1}) + \frac{1}{2\eta_t} \|\bar{w}_{t-1} - w_{t-1}\|^2 - \frac{1/\eta_t - \nu}{2} \left\| \frac{1}{I} \sum_{\xi \in I_t} w_t^{(\xi)} - \bar{w}_{t-1} \right\|^2 + G^2 \eta_t$$

$$= \frac{1}{I} \sum_{\xi \in I_t} R_{B_t^{(\xi)}}^{(\xi)}(\bar{w}_{t-1}) + \frac{1}{2\eta_t} \|\bar{w}_{t-1} - w_{t-1}\|^2 - \frac{1/\eta_t - \nu}{2} \|w_t - \bar{w}_{t-1}\|^2 + G^2 \eta_t,$$

Conditioned on $\mathcal{F}_{t-1}$, taking expectation (w.r.t. both the randomness of device sampling and data sampling introduced associated with the iteration step $t$) over both sides of the above inequality leads

to the following:

$$\mathbb{E}\left[\bar{R}(w_{t-1}) + \frac{1}{2\eta_t}\|w_t - w_{t-1}\|^2 \mid \mathcal{F}_{t-1}\right]$$

$$=\mathbb{E}\left[\frac{1}{I}\sum_{\xi \in I_t} R^{(\xi)}(w_{t-1}) + \frac{1}{2\eta_t}\|w_t - w_{t-1}\|^2 \mid \mathcal{F}_{t-1}\right]$$

$$=\mathbb{E}\left[\frac{1}{I}\sum_{\xi \in I_t} R^{(\xi)}_{B_t^{(\xi)}}(w_{t-1}) + \frac{1}{2\eta_t}\|w_t - w_{t-1}\|^2 \mid \mathcal{F}_{t-1}\right]$$

$$\leq\mathbb{E}\left[\frac{1}{I}\sum_{\xi \in I_t} R^{(\xi)}_{B_t^{(\xi)}}(\bar{w}_{t-1}) + \frac{1}{2\eta_t}\|\bar{w}_{t-1} - w_{t-1}\|^2 - \frac{1/\eta_t - \nu}{2}\|w_t - \bar{w}_{t-1}\|^2 + G^2\eta_t \mid \mathcal{F}_{t-1}\right]$$

$$=\mathbb{E}\left[\bar{R}(\bar{w}_{t-1}) + \frac{1}{2\eta_t}\|\bar{w}_{t-1} - w_{t-1}\|^2 - \frac{1/\eta_t - \nu}{2}\|w_t - \bar{w}_{t-1}\|^2 + G^2\eta_t \mid \mathcal{F}_{t-1}\right].$$

Based the above inequality and by applying Lemma 12 again we can show that

$$\mathbb{E}\left[\bar{R}(w_t) + \frac{1}{2\eta_t}\|w_t - w_{t-1}\|^2 \mid \mathcal{F}_{t-1}\right]$$

$$=\mathbb{E}\left[\bar{R}(w_{t-1}) + \frac{1}{2\eta_t}\|w_t - w_{t-1}\|^2 + \bar{R}(w_t) - \bar{R}(w_{t-1}) \mid \mathcal{F}_{t-1}\right] \tag{20}$$

$$\leq\mathbb{E}\left[\bar{R}(w_{t-1}) + \frac{1}{2\eta_t}\|w_t - w_{t-1}\|^2 + G\|w_t - w_{t-1}\| \mid \mathcal{F}_{t-1}\right]$$

$$\leq\mathbb{E}\left[\bar{R}(\bar{w}_{t-1}) + \frac{1}{2\eta_t}\|\bar{w}_{t-1} - w_{t-1}\|^2 - \frac{1/\eta_t - \nu}{2}\|w_t - \bar{w}_{t-1}\|^2 + 2G^2\eta_t \mid \mathcal{F}_{t-1}\right],$$

where in the last inequality we have used $\|w_t - w_{t-1}\| \leq \frac{1}{I}\sum_{\xi \in I_t}\|w_t^{(\xi)} - w_{t-1}\| \leq G\eta_t$ due to triangle inequality and Lemma 12.

Since $\bar{R}$ is also $\nu$-weakly convex, invoking Lemma 7 to $\bar{w}_{t-1} = \text{prox}_{\rho\bar{R}_{\text{erm}}}(w_{t-1})$ yields

$$\bar{R}(\bar{w}_{t-1}) + \frac{1}{2\rho}\|\bar{w}_{t-1} - w_{t-1}\|^2 \leq \bar{R}(w_t) + \frac{1}{2\rho}\|w_t - w_{t-1}\|^2 - \frac{1/\rho - \nu}{2}\|\bar{w}_{t-1} - w_t\|^2,$$

which immediately gives the following conditioned expectation bound:

$$\mathbb{E}\left[\bar{R}(\bar{w}_{t-1}) + \frac{1}{2\rho}\|\bar{w}_{t-1} - w_{t-1}\|^2 \mid \mathcal{F}_{t-1}\right]$$

$$\leq\mathbb{E}\left[\bar{R}(w_t) + \frac{1}{2\rho}\|w_t - w_{t-1}\|^2 - \frac{1/\rho - \nu}{2}\|\bar{w}_{t-1} - w_t\|^2 \mid \mathcal{F}_{t-1}\right]. \tag{21}$$

By summing up (20) and (21) we get

$$\mathbb{E}\left[\frac{1/\eta_t - 1/\rho}{2}\|w_t - w_{t-1}\|^2 \mid \mathcal{F}_{t-1}\right]$$

$$\leq\mathbb{E}\left[\frac{1/\eta_t - 1/\rho}{2}\|\bar{w}_{t-1} - w_{t-1}\|^2 - \frac{1/\eta_t + 1/\rho - 2\nu}{2}\|\bar{w}_{t-1} - w_t\|^2 + 2G^2\eta_t \mid \mathcal{F}_{t-1}\right].$$

Since by assumption $\eta_t \leq \rho$, rearranging the terms in the above yields

$$\mathbb{E}\left[\|w_t - \bar{w}_{t-1}\|^2 \mid \mathcal{F}_{t-1}\right]$$

$$\leq\frac{1/\eta_t - 1/\rho}{1/\eta_t + 1/\rho - 2\nu}\|\bar{w}_{t-1} - w_{t-1}\|^2 + \frac{4G^2\eta_t}{1/\eta_t + 1/\rho - 2\nu}$$

$$\leq\|\bar{w}_{t-1} - w_{t-1}\|^2 - \frac{2(1/\rho - \nu)}{1/\eta_t + 1/\rho - 2\nu}\|\bar{w}_{t-1} - w_{t-1}\|^2 + \frac{4G^2\eta_t}{1/\eta_t + 1/\rho - 2\nu}.$$

Then based on the above and the definition of Moreau envelope we can show that

$$\mathbb{E}\left[\bar{R}_\rho(w_t) \mid \mathcal{F}_{t-1}\right]$$

$$=\mathbb{E}\left[\bar{R}(\bar{w}_t) + \frac{1}{2\rho}\|\bar{w}_t - w_t\|^2 \mid \mathcal{F}_{t-1}\right]$$

$$\leq\mathbb{E}\left[\bar{R}(\bar{w}_{t-1}) + \frac{1}{2\rho}\|\bar{w}_{t-1} - w_t\|^2 \mid \mathcal{F}_{t-1}\right]$$

$$=\bar{R}(\bar{w}_{t-1}) + \frac{1}{2\rho}\mathbb{E}\left[\|\bar{w}_{t-1} - w_t\|^2 \mid \mathcal{F}_{t-1}\right]$$

$$\leq\bar{R}(\bar{w}_{t-1}) + \frac{1}{2\rho}\|\bar{w}_{t-1} - w_{t-1}\|^2 - \frac{(1/\rho - \nu)/\rho}{1/\eta_t + 1/\rho - 2\nu}\|\bar{w}_{t-1} - w_{t-1}\|^2 + \frac{2G^2\eta_t/\rho}{1/\eta_t + 1/\rho - 2\nu}$$

$$=\bar{R}_\rho(w_{t-1}) - \frac{(1/\rho - \nu)/\rho}{1/\eta_t + 1/\rho - 2\nu}\|\bar{w}_{t-1} - w_{t-1}\|^2 + \frac{2G^2\eta_t/\rho}{1/\eta_t + 1/\rho - 2\nu}$$

$$=\bar{R}_\rho(w_{t-1}) - \frac{1 - \rho\nu}{1/\eta_t + 1/\rho - 2\nu}\left\|\nabla\bar{R}_\rho(w_{t-1})\right\|^2 + \frac{2G^2\eta_t/\rho}{1/\eta_t + 1/\rho - 2\nu},$$

where in the last equality we have used the identity $\|\bar{w}_{t-1} - w_{t-1}\|^2 = \rho^2\left\|\nabla\bar{R}_\rho(w_{t-1})\right\|^2$ (see, e.g., Davis and Drusvyatskiy, 2019). By rearranging the terms in the above and taking expectation over $\mathcal{F}_{t-1}$ we obtain that

$$\frac{1 - \rho\nu}{1/\eta_t + 1/\rho - 2\nu}\mathbb{E}\left[\left\|\nabla\bar{R}_\rho(w_{t-1})\right\|^2\right] \leq \mathbb{E}\left[\bar{R}_\rho(w_{t-1})\right] - \mathbb{E}\left[\bar{R}_\rho(w_t)\right] + \frac{2G^2\eta_t/\rho}{1/\eta_t + 1/\rho - 2\nu}.$$

Averaging the above over $t = 1, ..., T$ yields

$$\frac{1}{T}\sum_{t=0}^{T-1}\mathbb{E}\left[\left\|\nabla\bar{R}_\rho(w_t)\right\|^2\right] \leq \frac{1/\eta_t + 1/\rho - 2\nu}{T(1 - \rho\nu)}\mathbb{E}\left[\bar{R}_\rho(w_0) - \bar{R}_\rho(w_T)\right] + \frac{2G^2\eta_t}{\rho(1 - \rho\nu)}$$

$$\leq \frac{1/\eta_t + 1/\rho - 2\nu}{T(1 - \rho\nu)}\bar{\Delta}_\rho^{(0)} + \frac{2G^2\eta_t}{\rho(1 - \rho\nu)}$$

$$= \frac{(1 - 2\rho\nu)\bar{\Delta}_\rho^{(0)}}{T\rho(1 - \rho\nu)} + \frac{\bar{\Delta}_\rho^{(0)}}{\eta_t T(1 - \rho\nu)} + \frac{2G^2\eta_t}{\rho(1 - \rho\nu)}$$

$$\leq \frac{\bar{\Delta}_\rho^{(0)}}{T\rho} + \frac{2\bar{\Delta}_\rho^{(0)}}{\eta_t T} + \frac{4G^2\eta_t}{\rho}$$

$$= \frac{\bar{\Delta}_\rho^{(0)}}{T\rho} + \frac{2\bar{\Delta}_\rho^{(0)} + 4G^2\rho}{\rho\sqrt{T}},$$

where in the last but one inequality we have used $\rho < \frac{1}{2\nu}$, and in the last inequality we have used the choice of $\eta_t \equiv \frac{\rho}{\sqrt{T}}$. The desired bound follows by preserving the dominant terms in the above bound and appealing to the definition of $t^*$. $\square$

## D    Proofs of Preliminary Lemmas

Here we provide the proofs of some auxiliary lemmas introduced in Appendix A.

### D.1    Proof of Lemma 1

*Proof.* Let $w_S^* = \arg\min_{w \in \mathbb{R}^p} R_S^r(w)$. Based on the strong convexity of $R_S^r(w_S)$ we can see that

$$\frac{\lambda}{2}\|w_S - w_S^*\|^2 \leq R_S^r(w_S) - R_S^r(w_S^*) \leq \varepsilon_t,$$

which directly implies $\|w_S - w_S^*\| \leq \sqrt{\frac{2\varepsilon_t}{\lambda}}$. Let us consider a sample set $S^{(i)}$ which is identical to $S$ except that one of the $z_i$ is replaced by another random sample $z_i'$. Denote

$w_{S^{(i)}}^* = \arg\min_{w \in \mathbb{R}^p} R_{S^{(i)}}^r(w)$. Then we can show that

$$
\begin{aligned}
& R_S^r(w_{S^{(i)}}^*) - R_S^r(w_S^*) \\
=& \frac{1}{N} \sum_{j \neq i} \left( \ell(w_{S^{(i)}}^*; z_j) - \ell(w_S^*; z_j) \right) + \frac{1}{N} \left( \ell(w_{S^{(i)}}^*; z_i) - \ell(w_S^*; z_i) \right) + r(w_{S^{(i)}}^*) - r(w_S^*) \\
=& R_{S^{(i)}}^r(w_{S^{(i)}}^*) - R_{S^{(i)}}^r(w_S^*) + \frac{1}{N} \left( \ell(w_{S^{(i)}}^*; z_i) - \ell(w_S^*; z_i) \right) - \frac{1}{N} \left( \ell(w_{S^{(i)}}^*; z_i') - \ell(w_S^*; z_i') \right) \\
\overset{\zeta_1}{\leq}& \frac{1}{N} \left( \ell(w_{S^{(i)}}^*; z_i) - \ell(w_S^*; z_i) \right) - \frac{1}{N} \left( \ell(w_{S^{(i)}}^*; z_i') - \ell(w_S^*; z_i') \right) \\
\overset{\zeta_2}{\leq}& \frac{2G}{N} \| w_{S^{(i)}}^* - w_S^* \|,
\end{aligned}
$$

where "$\zeta_1$" follows from the optimality of $w_{S^{(i)}}$ and "$\zeta_2$" is due to the Lipschitz continuity of loss. The strong convexity of $R_S^r$ implies

$$
R_S^r(w_{S^{(i)}}^*) - R_S^r(w_S^*) \geq \frac{\lambda}{2} \| w_{S^{(i)}}^* - w_S^* \|^2.
$$

Combining the preceding two inequalities yields

$$
\| w_{S^{(i)}}^* - w_S^* \| \leq \frac{4G}{\lambda N}.
$$

Therefore by triangle inequality and the above bounds we get

$$
\begin{aligned}
\| w_{S^{(i)}} - w_S \| =& \| w_{S^{(i)}} - w_{S^{(i)}}^* + w_{S^{(i)}}^* - w_S^* + w_S^* - w_S \| \\
\leq& \| w_{S^{(i)}} - w_{S^{(i)}}^* \| + \| w_{S^{(i)}}^* - w_S^* \| + \| w_S^* - w_S \| \\
\leq& \frac{4G}{\lambda N} + 2\sqrt{\frac{2\varepsilon_t}{\lambda}}
\end{aligned}
$$

which implies the desired uniform stability as the above holds for any pair of $S^{(i)}$ and $S$. □

## D.2  Proof of Lemma 3

*Proof.* Let us consider a sample set $S^{(i)}$ which is identical to $S$ except that one of the $Z_i$ is replaced by another random sample $Z_i'$. Since $S$ and $S^{(i)}$ are both i.i.d. samples of the data distribution. It follows that

$$
\mathbb{E}_S \left[ \nabla R(A(S)) \right] = \mathbb{E}_{S^{(i)}} \left[ \nabla R(A(S^{(i)})) \right] = \mathbb{E}_{S^{(i)} \cup \{Z_i\}} \left[ \nabla \ell(A(S^{(i)})); Z_i) \right] = \mathbb{E}_{S \cup \{Z_i'\}} \left[ \nabla \ell(A(S^{(i)}); Z_i) \right].
$$

Since the above holds for all $i = 1, ..., N$, by averaging the above equality over the training data we obtain that

$$
\mathbb{E}_S \left[ \nabla R(A(S)) \right] = \frac{1}{N} \sum_{i=1}^N \mathbb{E}_{S \cup \{Z_i'\}} \left[ \nabla \ell(A(S^{(i)}); Z_i) \right]. \tag{22}
$$

Regarding the empirical case, by definition we have

$$
\mathbb{E}_S \left[ \nabla R_S(A(S)) \right] = \frac{1}{N} \sum_{i=1}^N \mathbb{E}_S \left[ \nabla \ell(A(S); Z_i) \right] = \frac{1}{N} \sum_{i=1}^N \mathbb{E}_{S \cup \{Z_i'\}} \left[ \nabla \ell(A(S); Z_i) \right].
$$

Combining the preceding two equalities gives that

$$
\begin{aligned}
\| \mathbb{E}_S \left[ \nabla R(A(S)) - \nabla R_S(A(S)) \right] \| =& \left\| \frac{1}{N} \sum_{i=1}^N \mathbb{E}_{S \cup \{Z_i'\}} \left[ \nabla \ell(A(S^{(i)}); Z_i) - \nabla \ell(A(S); Z_i) \right] \right\| \\
\leq& L \left\| A(S^{(i)}) - A(S) \right\| \leq L\gamma,
\end{aligned}
$$

where we have used the uniform stability of $A$.

To prove the second inequality, again by smoothness of the loss function we have

$$
\left\| \nabla R(A(S)) - \nabla R(A(S^{(i)})) \right\| \leq L \left\| A(S) - A(S^{(i)}) \right\| \leq L\gamma.
$$

Then it follows from Lemma 2 that

$$
\mathbb{E}_S \left[ \| \nabla R(A(S)) - \mathbb{E}_S \left[ \nabla R_S(A(S)) \right] \|^2 \right] \leq L^2 \gamma^2 N.
$$

The proof is completed. □

# E  Preliminary Experimental Results

In this section, we carry out a preliminary experimental study to demonstrate the speed-up behavior of `FedMSPP` under varying minibatch sizes for achieving comparable test performances to `FedProx`. We also conventionally use `FedAvg` as a baseline algorithm for comparison.

## E.1  Data and Models

We compare the considered algorithms over the following three benchmark data sets popularly used for evaluating heterogenous FL approaches:

- The MNIST (LeCun et al., 1998) dataset of handwritten digits 0-9 is used for digit image classification with a two layer convolutional neural network (CNN). The model takes as input the images of size $28 \times 28$, and first performs a 2-layer ({1, 32, max-pooling}, {32, 64, max-pooling}) convolution followed by a fully connected (FC) layer. We use 63,000 images in which $90\%$ are for training and the rest for test. The data are distributed over 100 devices such that each device has samples of only 2 digits.

- The FEMNIST (Li et al., 2020b) dataset is a subset of the 62-class EMNIST (Cohen et al., 2017) database constructed by sub-sampling 10 lower case characters ('a'-'j'). We study the performances of the considered algorithms for character image classification using the same two layer CNN as used for MNIST, which takes as input the images of size $28 \times 28$. We use 55,050 images in which $90\%$ are for training and the rest for test. The data are distributed over 50 devices, each of which has samples of 3 characters.

- The Sent140 (Go et al., 2009) dataset of text sentiment analysis on tweets is used for evaluating the considered algorithms for sentiment classification. The model we use is a two layer LSTM binary classifier containing 256 hidden units followed by a densely-connected layer. The input is a sequence of 25 characters represented by a 300-dimensional GloVe embedding (Pennington et al., 2014) and the output is one character per training sample. We use for our experiment a total number of $21,546$ tweets from 261 twitter accounts, each of which corresponds to a device. The training/test sample split is $80\%$ versus $20\%$.

The statistics of the data and models in use are summarized in Table 2.

| Dataset | Model | # Devices | # Samples (Training) |
|---------|-------|-----------|----------------------|
| MNIST | 2-layer CNN | 100 | $63,000$ (56700) |
| FEMNIST | 2-layer CNN | 50 | 55050 (49545) |
| Sent140 | 2-layer LSTM | 261 | 21546 (17237) |

Table 2: Statistics of data and models used in the experiments.

## E.2  Implementation Details and Performance Metrics

We generally follow the instructions of Li et al. (2020b) for implementing `FedProx`, `FedMSPP` and `FedAvg`. More specifically, we use SGD as the local solver for `FedProx`, `FedMSPP` and `FedAvg`. For `FedMSPP`, we implement with three varying minibatch sizes on each data set as shortly reported in the next subsection about results. The hyper-parameters used in our implementation, such as number of communication rounds and number of local SGD epochs, are listed in Table 3.

| Hyper-parameter | MNIST | FEMNIST | Sent140 |
|-----------------|-------|---------|---------|
| #Communication rounds | 200 | 300 | 300 |
| #Local SGD epochs | 2 | 5 | 10 |
| Local SGD minibatch size | 567 | 512 | 100 |
| Local SGD learning rate | 0.25 | 0.06 | 0.1 |
| Strength of regularization $\mu_t$ | 0.1 | 0.1 | 0.001 |

Table 3: Hyper-parameter settings.

Since the chief goal of this empirical study is to illustrate the benefit of `FedMSPP` for speeding up the convergence of `FedProx`, we use the numbers of data points and communication rounds needed

for reaching the desired solution accuracy as performance metrics. The desired test accuracies are $\{80\%, 90\%, 95\%\}$ on MNIST, $\{80\%, 85\%, 91\%\}$ on FEMNIST, and $\{68\%, 70\%, 73\%\}$ on Sent140.

### E.3 Results

In Figure 1, we show the numbers of data samples accessed by the considered algorithms to reach comparable test accuracies. For FedMSPP, we test with minibatch sizes $\{81, 63, 10\}$ on MNIST, $\{128, 64, 16\}$ on FEMNIST, and $\{75, 50, 20\}$ on Sent140. From this set of results we can observe that:

- On all the three data sets in use, FedMSPP with varying minibatch sizes consistently needs significantly fewer samples than FedProx and FedAvg to reach the desired test accuracies.
- FedMSPP with smaller minibatch size tends to have better sample efficiency.

Figure 2 shows the corresponding rounds of communication needed to reach comparable test accuracies. From this group results we can see that in most cases, FedMSPP just needs slightly increased rounds of communication than FedProx and FedAvg to reach comparable generalization accuracy.

Overall, our numerical results confirm that FedMSPP can be served as a safe and computationally more efficient replacement to FedProx on the considered heterogenous FL tasks.

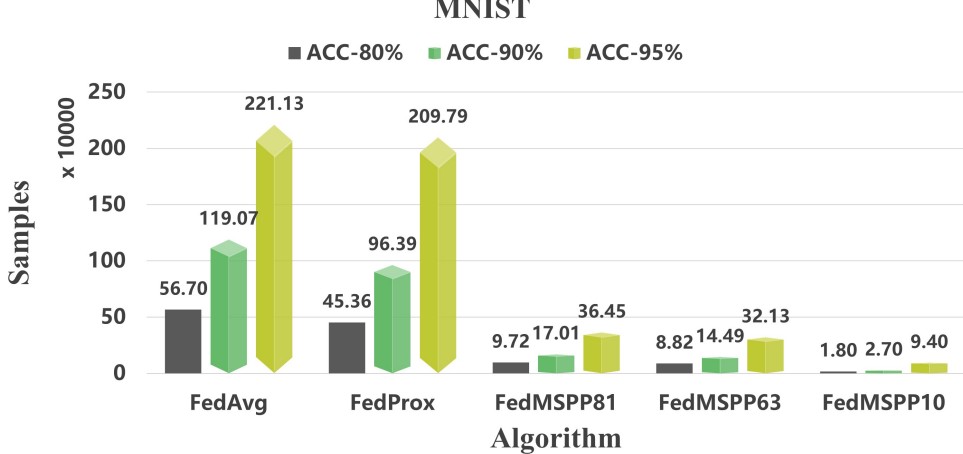

(a) MNIST: Numbers of data points needed to reach 80%, 90% and 95% test accuracies. For `FedMSPP`, we test with different minibatch sizes 81, 63, 10.

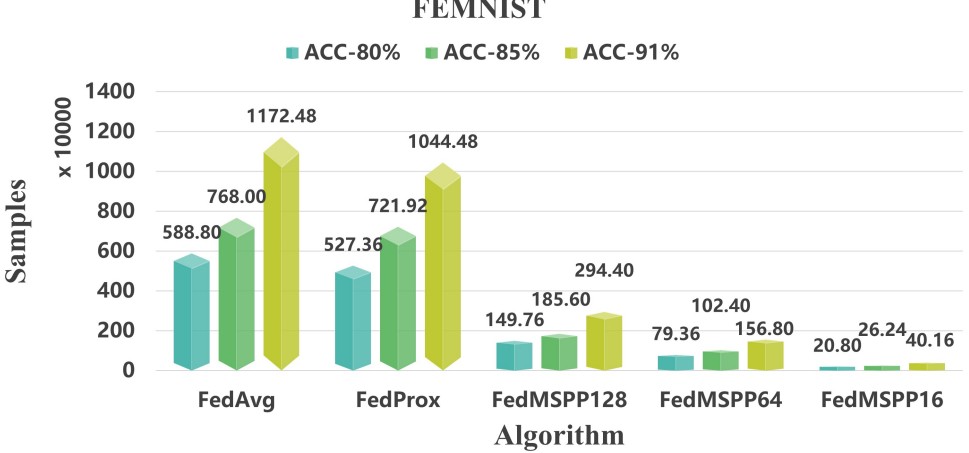

(b) FEMNIST: Numbers of data points needed to reach 80%, 85% and 91% test accuracies. For `FedMSPP`, we test with different minibatch sizes 128, 64, 16.

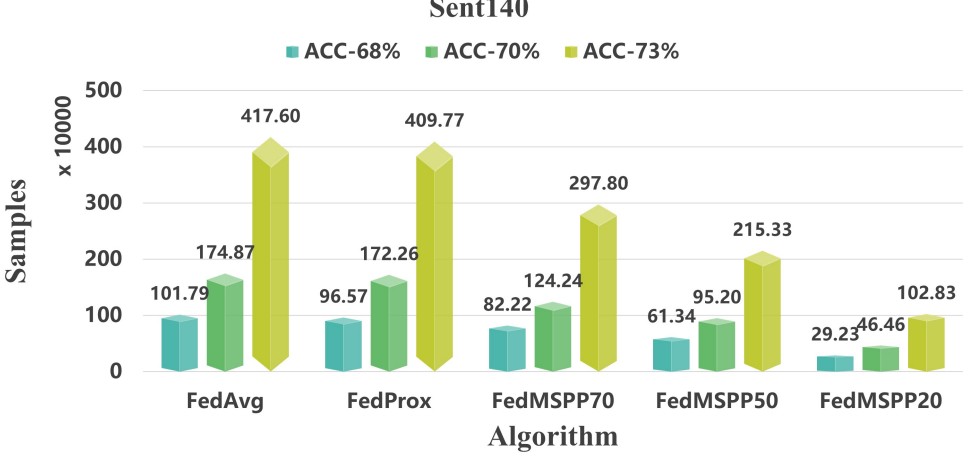

(c) Sent140: Numbers of data points needed to reach 68%, 70% and 73% test accuracies. For `FedMSPP`, we test with different minibatch sizes 70, 50, 20.

Figure 1: Comparison of numbers of data points accessed by the considered algorithms to reach varying desired test accuracies.

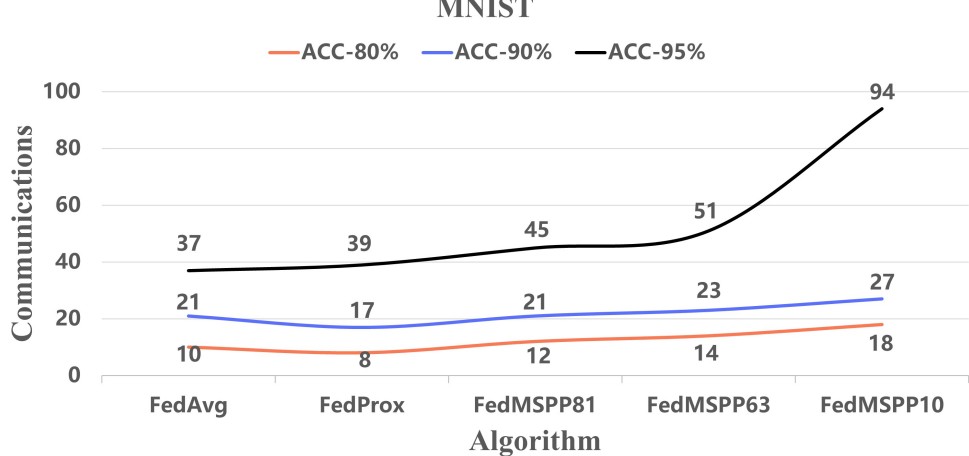

(a) MNIST: Rounds of communication needed to reach 80%, 90% and 95% test accuracies. For `FedMSPP`, we test with different minibatch sizes 81, 63, 10.

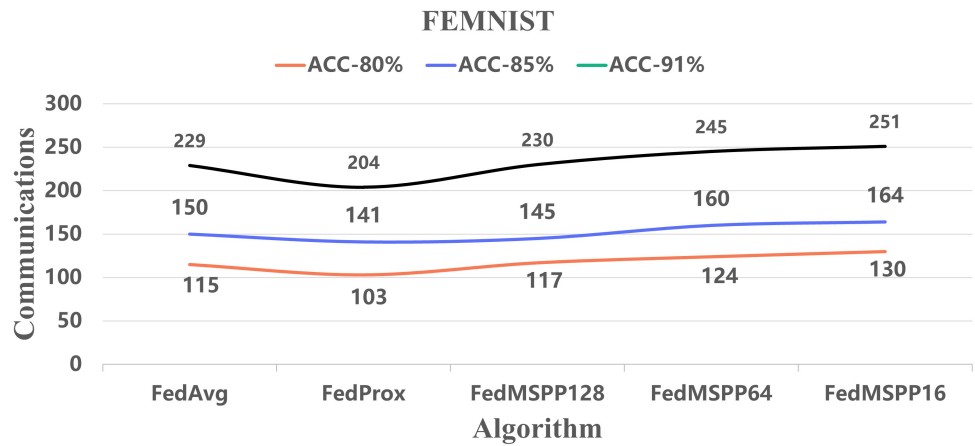

(b) FEMNIST: Rounds of communication needed to reach 80%, 85% and 91% test accuracies. For `FedMSPP`, we test with different minibatch sizes 128, 64, 16.

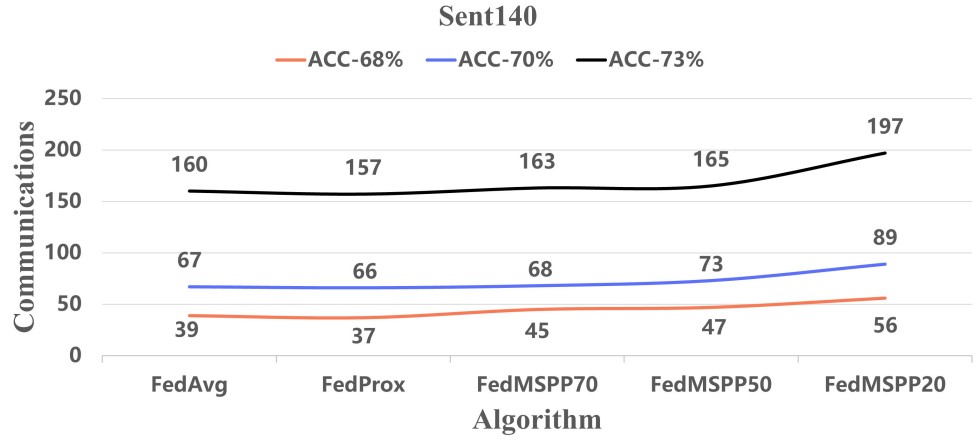

(c) Sent140: Rounds of communication needed to reach 68%, 70% and 73% test accuracies. For `FedMSPP`, we test with different minibatch sizes 70, 50, 20.

Figure 2: Comparison of rounds of communication needed by the considered algorithms to reach varying desired test accuracies.