# OpenReview forum: "On Convergence of FedProx: Local Dissimilarity Invariant Bounds, Non-smoothness and Beyond"
_NeurIPS.cc/2022/Conference — NeurIPS 2022 Accept_

### Official Review · Reviewer_TmSS · 2022-07-10

**Rating:** 6
**Confidence:** 3
**Soundness:** 3 good
**Presentation:** 3 good
**Contribution:** 3 good

**Summary:**

This is theoretical paper and it aims to study the convergence analysis of FedProx. The difference in this paper compared to existing works is that it does not assume the constraints on local dissimilarity, smoothness and the size of minibach. The novelty is to use algorithm stability theory to perform the analysis. In addition, the authors also provide a FedMSPP algorithm that is a minibatch stochastic extension of FedProx and perform its convergence analysis.

**Questions:**

1. In Theorem 1, is the result obtained by assuming 1 data sample point used per iteration?

2. What is the real novelty in terms of analysis, please indicate the key step that is different from the approach used in the literature?

3. In Theorems 1 and 2,, what is the reason to use t* instead of averaging all the terms?



**Limitations:**

The convergence rate can be loose. First, it can be worse than what was given in the literature. The authors used a few times that the obtained convergence rate is comparable with those in the literature. It is unclear this is because of the mathematical tools used in this paper or this is the result of relaxing the assumption on the local dissimilarity and/or smoothness. Second, as mentioned before, Theorems 2 and 4 can be very loose since it is not a function of I and/or d (the authors also mentioned this in the end of the paper). If the results are too loose, the contribution of this paper may be weakened.

**Strengths And Weaknesses:**

Strengths:

1. The authors claim that their theory is the first time local dissimilarity and smoothness are not necessary to guarantee the convergence of FedProx with reasonbale rates.

2. The authors propose FedMSPP algorithm and perform its convergence analysis.

Weaknesses:

1. The main novelty of this paper is a new analysis method without assuming local dissimilarity and smoothness. However, the authors failed to clearly explain the details of this novelty.

2. The obtained convergence rates can be loose. For example, in Theorem 2, the convergence rate is not a function of I, which seems not make sense. In Theorem 4, the convergence rate is not a function of I and d, which also seems not make sense.

---

> ### Author Response · Authors · 2022-08-02
> **Response to Reviewer TmSS**
>
> Thank you for the positive feedback and insightful comments. We sincerely hope that the main concerns raised in the review can be clarified by the following response.
>
> > **Your comment:** The main novelty of this paper is a new analysis method without assuming local dissimilarity and smoothness. However, the authors failed to clearly explain the details of this novelty.
>
> **Our response:** We would like to clarify that the novel ingredients of our analysis have indeed been elaborated on where appropriate in the main submission. Particularly, the non-trivial elements developed in the proofs of Theorem 2 and Theorem 3 are highlighted with details in the corresponding proof sketches provided along with these theorems.
>
> > **Your comment:** The obtained convergence rates can be loose. For example, in Theorem 2, the convergence rate is not a function of I, which seems not make sense. In Theorem 4, the convergence rate is not a function of I and d, which also seems not make sense.
>
> **Our response:** While not demonstrating any linear speedup with respect to device sampling ratio and local minibatch size, our results in Theorem 2 and Theorem 4 for the first time provide provable convergence guarantees for FedProx and FedMSPP for non-smooth and non-convex problems. We expect these breakthrough results would stimulate future research on non-smooth FL problems hopefully with improved rates of convergence.
>
> >**Your question:** In Theorem 1, is the result obtained by assuming 1 data sample point used per iteration?
>
> **Our response:** Actually, as expressed in Equation (3) that we just follow the venilla FedProx to use the *entire data set* on each device for implementing its local proximal point update oracle.
>
> > **Your question:** What is the real novelty in terms of analysis, please indicate the key step that is different from the approach used in the literature?
>
> **Our response:**
>
> 1. For Theorems 2 - 4 which represent the major breakthrough results of this work, we kindly refer the reviewer to the corresponding proof sketches in the main submission for the novel steps in analysis. In particular, the proof technique of Theorem 2 is inspired by the arguments by Davis and Drusvyatskiy [2019] developed for stochastic model-based algorithms, with several new elements along developed for handling the model averaging and partial participation mechanisms of FedProx. Regarding the proof of Theorem 3, a key novel ingredient is to show via an extended uniform stability arguments for gradients (see Lemma 3) that the averaged directions $d_t $ aligns well with the global gradient $\nabla \bar R (w_{t-1})$ in expectation (see Lemma 11).
>
> 2. As for Theorem 1, the proof is an adaptation of the standard arguments for smooth optimization to FedProx. While being more or less straightforward to derive, the result in Theorem 1 turns out to be valid under much weaker conditions than the best known for FedProx. Please see Remark 1 for the related discussions on the novelty and strength of Theorem 1.
>
> > **Your question:** In Theorems 1 and 2,, what is the reason to use $t^*$ instead of averaging all the terms?
>
> **Our response:** It is indeed optional to report the bounds in terms of the averaged gradient norm or the gradient norm at a specific $t^*$. The reason to choose the latter here is simply because we prefer to provide guarantees on a single output carefully chosen from the intermediate iterates, which is believed to be of practical interest.
>
> ## References:
>
> Damek Davis and Dmitriy Drusvyatskiy. Stochastic model-based minimization of weakly convex functions. *SIAM Journal on Optimization*, 29(1):207–239, 2019.

---

### Official Review · Reviewer_TUCL · 2022-07-11

**Rating:** 6
**Confidence:** 4
**Soundness:** 3 good
**Presentation:** 3 good
**Contribution:** 2 fair

**Summary:**

This paper studies the FedProx algorithm for nonconvex optimization. FedProx is a distributed version of the stochastic proximal point method that has found many applications in practice, and which generally seems to perform well in practice. FedProx was introduced and analyzed in [1], but the convergence rate given in [1] depends on a highly non-standard assumption, the strong growth condition, which this paper terms the (B, 0)-LGD assumption. The aim of this paper is to study the convergence of FedProx without this assumption; instead using the more-common assumption that each objective is Lipschitz. The authors give a convergence rate for smooth objectives (Theorem 1) that does not depend on the (B, 0)-LGD assumption, and a similar rate for non-smooth but weakly convex problems (Theorem 2)-- in addition, they develop minibatch variants of the algorithm and give convergence rates for it in Theorem 3 (smooth) and Theorem 4 (non-smooth).

[1] Tian Li, Anit Kumar Sahu, Manzil Zaheer, Maziar Sanjabi, Ameet Talwalkar, Virginia Smith. Federated Optimization in Heterogeneous Networks, mlsys.org (2020).


**Questions:**

- Can your analysis be extended to use bounded variance, rather than the assumption that each function is Lipschitz?


**Limitations:**

See the previous sections for limitations. Potential negative societal impacts are not applicable here.


**Strengths And Weaknesses:**

Main points:
- The convergence rate given by Theorem 1 is a welcome improvement upon the prior convergence result given in [1], as it does not require the (B, 0) condition and thus can be used in more general circumstances. On the other hand, this is quite limited by the alternative assumption used: the requirement that each function is Lipschitz itself is a control on the gradient dissimilarity, and does not allow the rate to show improvement with lower function dissimilarity.
- The rate for non-smooth objectives given in Theorem 2 is also novel and interesting in its own right, as it adds to the arsenal of algorithms that are analyzed in the setting of nonconvex but weakly convex objectives. While the authors state that it is the first convergence rate for FL algorithms, the rate given by [2] for SGD can be easily applied to federated learning, by seeing the sampling of stochastic gradients as sampling clients.
- The minibatch variants of the method are a nice addition, even though Theorem 4 does not show that minibatching results in any improvement for the algorithm under non-smoothness.
- None of the rates is asymptotically better than ordinary distributed SGD. This is quite puzzling, as FedProx requires a much stronger computational oracle and seems to perform well in practice. However, this could be inherent to the algorithm, rather than a product of a bad analysis. Without a lower bound for FedProx, we can not tell.
- The novelty of the proof for non-smooth problems is quite limited, especially since the proof shows no benefit from minibatching-- it follows the same outline as in [2].

I believe that this paper improves over the existing analysis and is a good addition to the literature, despite the rate itself not being very good compared to the baseline SGD. While it relaxes the (B, 0) condition required in [1], it also adds the Lipschitz condition that is more restrictive compared to prior work. So while this paper has some good, it is also more restrictive in other parts.

[1] Tian Li, Anit Kumar Sahu, Manzil Zaheer, Maziar Sanjabi, Ameet Talwalkar, Virginia Smith. Federated Optimization in Heterogeneous Networks, mlsys.org (2020).
[2] Damek Davis, Dmitriy Drusvyatskiy Stochastic Model-Based Minimization of Weakly Convex Functions, arXiv:1803.06523 (2018).

---

> ### Author Response · Authors · 2022-08-02
> **Response to Reviewer TUCL**
>
> Thank you for your insightful review and positive evaluation of our work. We sincerely hope the main concerns can be addressed satisfactorily by the following clarification.
>
> > **Your comment:** On the other hand, this is quite limited by the alternative assumption used: the requirement that each function is Lipschitz itself is a control on the gradient dissimilarity, and does not allow the rate to show improvement with lower function dissimilarity.
>
> **Our response:** We agree with the reviewer that the assumed Lipschitz-loss (or bounded gradient) condition implies that the local objective gradients cannot be too dissimilar. We have explicitly discussed this point in Remark 3 of the original submission. In the meanwhile, we would also like to stress that the Lipschitz-loss condition has indeed been commonly considered in the existing analysis of FL algorithms [see, e.g., Smith et al. 2017, Li et al. 2020, Zhang et al. 2020]. Some popular examples of Lipschitz loss include the logistic loss, hinge loss, Huber loss, and absolute loss, to name a few.
>
> > **Your comment:** While the authors state that it is the first convergence rate for FL algorithms, the rate given by [2] for SGD can be easily applied to federated learning, by seeing the sampling of stochastic gradients as sampling clients.
>
> **Our response:** It is indeed a very interesting point to extend the rate for non-smooth SGD [2, Theorem 4.3] to federated learning. We agree that the rate can be easily extended to the FL regime with single-round local SGD update. It seems however that for FL with multiple rounds of local SGD update, the suggested transformation of the result by [2] would be less straightforward.
>
> > **Your comment:** None of the rates is asymptotically better than ordinary distributed SGD. This is quite puzzling, as FedProx requires a much stronger computational oracle and seems to perform well in practice.
>
> **Our response:** We would like to highlight that compared to the local SGD update, the local (stochastic) proximal point update oracle used by FedProx is well known for being more robust and adaptive to the choice of step-sizes for iteration, but not necessarily faster in convergence rate (Asi and Duchi, 2019, Davis and Drusvyatskiy, 2019, Deng and Gao, 2021). It is conjectured that the improved robustness and stability of local update oracle should be beneficial for the global performance of FedProx both in theory and practice. We leave the full understanding of this issue for future investigation.
>
> > **Your question:** Can your analysis be extended to use bounded variance, rather than the assumption that each function is Lipschitz?
>
> **Our response:** It is a thrilling question to extend our analysis to the bounded gradient variance regime, which we are still trying to figure out ways to resolve. For now, our algorithmic stability theory inspired technique does not allow us to sidestep the Lipschitz-loss condition.
>
> ## References:
>
> Virginia Smith, Chao-Kai Chiang, Maziar Sanjabi, and Ameet Talwalkar. Federated multi-task learning. *NIPS*, 2017.
>
> Tian Li, Anit Kumar Sahu, Manzil Zaheer, Maziar Sanjabi, Ameet Talwalkar, and Virginia Smith. Federated optimization in heterogeneous networks. *MLSys*, 2:429–450, 2020.
>
> Xinwei Zhang, Mingyi Hong, Sairaj Dhople, Wotao Yin, and Yang Liu. Fedpd: A federated learning framework with optimal rates and adaptivity to non-iid data. *arXiv:2005.11418*, 2020.
>
> Hilal Asi and John C Duchi. Stochastic (approximate) proximal point methods: Convergence, optimality, and adaptivity. *SIAM Journal on Optimization*, 29(3):2257–2290, 2019.
>
> Damek Davis and Dmitriy Drusvyatskiy. Stochastic model-based minimization of weakly convex functions. *SIAM Journal on Optimization*, 29(1):207–239, 2019.
>
> Qi Deng and Wenzhi Gao. Minibatch and momentum model-based methods for stochastic non-smooth non-convex optimization. *NeurIPS*, 2021.

---

> > ### Comment · Reviewer_TUCL · 2022-08-08
> > **Response to authors**
> >
> > 1. On the bounded gradients condition: I know that you discussed it in the main work, and I know that it is common in some of the literature. I still don't see that as justifying this assumption. If this was the first analysis of the algorithm, it'd be justifiable. However, it is not. If the main point of your work is to get rid of an existing local dissimilarity condition that is too stringent, you shouldn't replace it by another dissimilarity condition that is also too stringent. And while I know that these losses are Lipschitz, they are never used in isolation. They're usually composed with another function in a way that makes the training objective non-Lipschitz.
> >
> > 2. I agree that multiple rounds of SGD locally are not easily covered by [4], however I then ask you to compare your result to [4] and to not state that this is the first analysis of FL for non-smooth non-convex objectives, because as you acknowledge this already exists.
> >
> > Overall, I am still really on the fence about this. I leave my score as it is.

---

> > > ### Author Response · Authors · 2022-08-09
> > > **Thank you for the additional comments**
> > >
> > > We sincerely thank the reviewer for providing the constructive comments. Per your suggestion, we are glad to highlight in the next draft the analysis of (naive) federated non-smooth SGD implied by the result of Davis and Drusvyatskiy (2019).

---

### Official Review · Reviewer_MGVY · 2022-07-13

**Rating:** 9
**Confidence:** 4
**Soundness:** 4 excellent
**Presentation:** 4 excellent
**Contribution:** 3 good

**Summary:**

In this paper, the authors study FedProx with an improved convergence analysis without the unrealistic assumption of local gradient dissimilarity. Instead, they introduce a novel local dissimilarity invariant convergence theory for FedProx. FedProx for nonsmooth optimization and its minibatch stochastic extension are also studied with convergence guarantees and complexity bounds.


**Questions:**

- Typos:
	- Line 244: independent to → independent of

**Limitations:**

The authors have adequately addressed the limitations of their work.

**Strengths And Weaknesses:**

Strengths:
- This work gets rid of the usual yet unrealistic assumption of local gradient dissimilarity in the original analysis of FedProx
- Both *smooth and nonconvex* and *nonsmooth and weakly convex* cases are studied with convergence rates
- Minibatch stochastic extension of FedProx is also studied with convergence guarantees, which also enjoys linear speedup in terms of minibatch size and partial participation ratio

Weaknesses:
- Experimental results could be preliminary

---

> ### Author Response · Authors · 2022-08-02
> **Response to Reviewer MGVY**
>
> Thank you for your insightful review and appreciation of our work. The typos mentioned have been fixed in the updated draft. Thanks!

---

### Official Review · Reviewer_Cx6F · 2022-07-15

**Rating:** 6
**Confidence:** 4
**Soundness:** 3 good
**Presentation:** 4 excellent
**Contribution:** 3 good

**Summary:**

This paper studies convergence of a previously proposed algorithm called FedProx. FedProx has been fairly popular in the FL community but its analysis is pointed out to depend on unrealistic assumptions, namely, on bounded local gradient dissimilarity assumption. In this work, the authors propose an analysis that doesn't need this assumption. Moreover, they also study convergence of FedProx when the objective is non-smooth. Finally, they propose a variant of FedProx called FedMSPP, which uses proximal-point update in its inner loop. They prove convergence of FedMSPP and provide some numerical results for this method in the Appendix.

# After rebuttal
I decided to increase the score from 3 to 6 since the technical issue has been fixed (I also increased "Soundness" rating from 1 to 3). I still think that a more interesting story could be presented, with at least some result saying why FedProx would be a good method.

**Questions:**

### Main questions
1. The proof of Lemma 5 seems to use $L$-smoothness of $Q_{\mathrm{erm}}^{(m)}$ (when the gradient norm is upper bounded by functional gap). However, $Q_{\mathrm{erm}}^{(m)}(w; w_{t-1})=R_{\mathrm{erm}}^{(m)}(w) + \frac{1}{2\eta_t}\Vert w-w_{t-1}\Vert^2$ seems to be ($L+\frac{1}{\eta_t}$)-smooth. For small $\eta_t$, this seems to make a big difference. As far as I can see, this may change the result of Lemma 5 to have a constant error, in which case the theory might break down.
2. Could the authors comment on why FedProx would be better than simply computing a single gradient $\nabla R_{\mathrm{erm}}^{(\xi)}(w)$ on each client $\xi\in I_t$? At least in the smooth case, it seems to me that there is no benefit from computing the prox. I think it makes sense to at least mention this in the "Limitations" section.
3. The proof of Lemma 4 is problematic. First of all, you take expectation with respect to sampling $I_t$, but then you write that
$\mathbb{E}\left[\frac{1}{|I_t|}\sum_{\xi\in I_t}d_t^{(\xi)} \right] = \frac{1}{|I_t|}\sum_{\xi\in I_t}\mathbb{E}\left[d_t^{(\xi)} \right].$
The problem here is that $I_t$ is stochastic so you cannot put it outisde expectation. The claim itself is correct, so this is just a minor comment.
You also say that "By uniform without-replacement sampling strategy", but then you rely "the independence among the indices in $I_t$". The issue here is that sampling without replacement does not give independent indices. This can be easily fixed, for instance, see Lemma 1 in "Random Reshuffling: Simple Analysis with Vast Improvements" for the derivation of variance. Alternatively, you can just assume uniform sampling (which would probably make less sense since we talk about client sampling).

### Typos:
Page 2: partial participants (participation) of devices
Page 5, Table 1: functions. The involved of quantities (should be no "of")
Page 9: At nutshell (in a nutshell)
Page 9: weak convex (weakly convex)
Page 16, line 561: (Proof) of Theorem 1. "Proof" is also missing on page 18, line 592; page 23, line 657; page 25, line 682.

**Limitations:**

The limitations were discussed well. I would be also interested in seeing a discussion on the utility of computing prox instead of computing gradients.

**Strengths And Weaknesses:**

## Strengths
1. The idea to study FedProx without bounded dissimilarity makes perfect sense to me. The method is quite popular in FL, where bounded dissimilarity is not always a reasonable assumption since clients data can be very heterogeneous.
2. The obtained complexities look good compared to that of other methods (see Table 1). The theory covers partial participation and shows improvement in terms of client batch size in case of smooth functions.
3. The work is well-written and the main contributions and ideas are stated clearly.

## Weaknesses
1. I do not understand one step in the proof of Lemma 5 (see "Questions" section of my review). This seems to be of fundamental importance, so if my question is not addressed, I would vote for rejection of the paper. On the other hand, if I miss something, I will be willing to drastically change the assigned score.
2. As far as I can see, there is no improvement from performing more than a single local step. This is a common issue with FL methods, but it is still disappointing.

---

> ### Author Response · Authors · 2022-08-02
> **Response to Reviewer Cx6F**
>
> Thank you for your insightful review and highly constructive comments for improvement. We sincerely hope that the main concerns raised in the review can been addressed satisfactorily in the following response.
>
> > **Your comment:** I do not understand one step in the proof of Lemma 5 (see "Questions" section of my review). This seems to be of fundamental importance, so if my question is not addressed, I would vote for rejection of the paper. On the other hand, if I miss something, I will be willing to drastically change the assigned score.
>
> **Our response:** Thanks for pointing out this minor flaw of Lemma 5 which can be easily fixed without impairing the strength of our main results. It is indeed true that the smoothness coefficient should be updated as $L+\eta_t^{-1}$ on the RHS of the bound. In the meanwhile, such a change of the multiplier of the sub-optimality $\varepsilon_t$ still guarantees a time-decaying error provided that $\varepsilon_t$ approaches zero fast enough. In the end, the updated smoothness coefficient turns out to only yield a slightly more stringent condition on the sub-optimality parameter $\varepsilon_t$ in Theorem 1. Moreprecisely, the condition is now updated as $\varepsilon_t \le \min  \left(\frac{2L^2G^2\eta_t^3}{I^2(L\eta_t+1)}, \frac{G^2\eta_t}{2I(L\eta_t+1)} \right)$. Similar minor modifications also apply to Lemma 9 and Theorem 3. Please see the updated draft for the detailed changes.
>
> > **Your comment:** Could the authors comment on why FedProx would be better than simply computing a single gradient? At least in the smooth case, it seems to me that there is no benefit from computing the prox. I think it makes sense to at least mention this in the "Limitations" section.
>
> **Our response:** Concerning the benefit of FedProx over SGD-type FL approaches, we would like to highlight the following aspects:
>
> 1. Our main results reveal that with minimal modifications, FedProx with local minibatch stochastic proximal point update enjoys favorable convergence rates that are 1) invariant to local dissimilarity, 2) applicable to smooth or non-smooth problems, and 3) scaling linearly with respect to local minibatch size and device sampling ratio for smooth problems. To our knowledge, it has not yet been possible to have all these appealing properties simultaneously guaranteed for a single existing FL framework.
>
> 2. More specifically compared to the local SGD update, the local stochastic proximal point (SPP) oracle used by FedProx has been widely recognized to be more robust and adaptive to the choice of step-sizes, but not necessarily faster in convergence rate, in the literature of stochastic model-based optimization (Asi and Duchi, 2019, Davis and Drusvyatskiy, 2019, Deng and Gao, 2021). It is reasonable to conjecture that similar robustness and stability of iteration should be inherited by FedProx for local and global update.
>
> 3. We agree with the reviewer that in the smooth-loss case, our current results do not reveal any substantial improvement of FedProx over SGD-type FL methods from the theoretical perspective. Per your suggestion, we have updated the draft by explicitly acknowledging this limitation at the end of the main paper.
>
> > **Your comment:** The proof of Lemma 4 is problematic.
>
> **Our response:** Thanks for pointing out the minor issues in the proof of Lemma 4 which we believe can be readily fixed. Indeed, since Lemma 4 is only used in the proof of Theorem 1 under the condition of $|I_t|\equiv I$, the randomness in $|I_t|$ would disappear if exactly the same condition is assumed in Lemma 4. Concerning the sampling strategy mentioned in the first line of proof, actually what we would really want to say is *uniform sampling*. Sorry for the confusion caused by this typo.
>
> > **Minor comments** on typos.
>
> **Our response:** Thanks for catching the typos which have been fixed in the updated draft.
>
> We sincerely hope that the reviewer might consider increasing the score if the main concerns are clarified by our response. In case that anything of concern is missing, please kindly let us know during the reviewer-author discussion phase and we will gladly update the response.
>
> ## References:
>
> Hilal Asi and John C Duchi. Stochastic (approximate) proximal point methods: Convergence, optimality, and adaptivity. *SIAM Journal on Optimization*, 29(3):2257–2290, 2019.
>
> Damek Davis and Dmitriy Drusvyatskiy. Stochastic model-based minimization of weakly convex functions. *SIAM Journal on Optimization*, 29(1):207–239, 2019.
>
> Qi Deng and Wenzhi Gao. Minibatch and momentum model-based methods for stochastic non-smooth non-convex optimization. *NeurIPS*, 2021.

---

> > ### Comment · Reviewer_Cx6F · 2022-08-04
> > **Thank you for fixing the issue**
> >
> > I thank the authors for addressing my concerns. It seems to me that the new results are correct. It also seems that the new (total) complexity is worse than before because $\varepsilon_t$ has to be smaller. The communication complexity seems to remain the same.
> > I also thank the authors for adding a discussion of the limitations when compared to Local SGD.
> >
> > I am satisfied with the authors' response and raise my score from 3 to 6 to acknowledge the improvements. I wish there was something to say about the benefits of using FedProx, which prevents me from raising the score further. At the same time, I wouldn't be unhappy if this paper gets accepted.

---

> > > ### Author Response · Authors · 2022-08-05
> > > **Thank you for your positive feedback**
> > >
> > > We sincerely thank the reviewer for upgrading the score and providing highly constructive comments which really helped us to improve the paper.

---

### Author Response · Authors · 2022-08-02
**General response**

We sincerely thank all the reviewers for their detailed and insightful comments on our work. We have carefully revised the manuscript based on the preliminary reviews. The following is a summary of major changes:

1. We have slightly modified the conditions on the local update oracle precision $\varepsilon_t$ in Theorem 1, Theorem 3 and Lemma 11.

2. L343-345: We explicitly acknowledge that in the smooth-loss case, our current results do not reveal any substantial improvement of FedProx over SGD-type FL methods from the theoretical perspective.

3. Lemmas 4, 5, 6, 9 are slightly modified in factors of bounds or proof arguments.

4. We have tried our best to fix typos and other minor issues.

The major changes are highlighted in red text. We hope that the given concerns have been addressed satisfactorily in the revised manuscript and the point-by-point responses to the reviewers' comments.

---

### Meta-Review · Area_Chair_PtG8 · 2022-08-25

**Recommendation:** Accept
**Confidence:** Certain

**Metareview:**

The reviewers generally agreed that the work makes a contribution towards weakening the assumptions when analyzing the well-known FedProx method. However, I strongly agree with one of the reviewers' comment that it is not satisfactory to justify an assumption (here, it is the bounded gradient) just by stating that it is common in the literature. It would be desirable, e.g., to give application scenarios that satisfy the assumption and explain the limitations/implications of the assumption. Please consider addressing this in your revision.

**Award:**

No

---

### Decision · Program_Chairs · 2022-09-14

Accept